# Revealing unexpected complex encoding but simple decoding mechanisms in motor cortex via separating behaviorally relevant neural signals

Yangang Li[1,2,3,4], Xinyun Zhu[1,2,3,4], Yu Qi[2,3,4,5]*, Yueming Wang[1,2,3,4,5]*†

[1]Qiushi Academy for Advanced Studies, Zhejiang University, Hangzhou, China; [2]Nanhu Brain-Computer Interface Institute, Hangzhou, China; [3]College of Computer Science and Technology, Zhejiang University, Hangzhou, China; [4]The State Key Lab of Brain-Machine Intelligence, Zhejiang University, Hangzhou, China; [5]Affiliated Mental Health Center & Hangzhou Seventh People's Hospital and the MOE Frontier Science Center for Brain Science and Brain-Machine Integration, Zhejiang University School of Medicine, Hangzhou, China

*For correspondence:
qiyu@zju.edu.cn (YQ);
ymingwang@zju.edu.cn (YW)

†Lead contact

**Abstract** In motor cortex, behaviorally relevant neural responses are entangled with irrelevant signals, which complicates the study of encoding and decoding mechanisms. It remains unclear whether behaviorally irrelevant signals could conceal some critical truth. One solution is to accurately separate behaviorally relevant and irrelevant signals at both single-neuron and single-trial levels, but this approach remains elusive due to the unknown ground truth of behaviorally relevant signals. Therefore, we propose a framework to define, extract, and validate behaviorally relevant signals. Analyzing separated signals in three monkeys performing different reaching tasks, we found neural responses previously considered to contain little information actually encode rich behavioral information in complex nonlinear ways. These responses are critical for neuronal redundancy and reveal movement behaviors occupy a higher-dimensional neural space than previously expected. Surprisingly, when incorporating often-ignored neural dimensions, behaviorally relevant signals can be decoded linearly with comparable performance to nonlinear decoding, suggesting linear readout may be performed in motor cortex. Our findings prompt that separating behaviorally relevant signals may help uncover more hidden cortical mechanisms.

## eLife assessment

This study presents a **useful** method for the extraction of behaviour-related activity from neural population recordings based on a specific deep learning architecture, a variational autoencoder. Although the authors performed thorough benchmarking of their method in the context of decoding behavioural variables, the evidence supporting claims about encoding is **incomplete** as the results may stem, in part, from the properties of the method itself.

## Introduction

Understanding how motor cortex encodes and decodes movement behaviors is a fundamental goal of neuroscience (*Kriegeskorte and Douglas, 2019*; *Saxena and Cunningham, 2019*). Here, we define behaviors as behavioral variables of interest measured within a given task, such as arm kinematics during a motor control task; we employ terms like 'behaviorally relevant' and 'behaviorally irrelevant'

only regarding such measured behavioral variables. However, achieving this goal faces significant challenges because behaviorally relevant neural responses are entangled with behaviorally irrelevant factors such as responses for other variables of no interest (*Fusi et al., 2016*; *Rigotti et al., 2013*) and ongoing noise (*Azouz and Gray, 1999*; *Faisal et al., 2008*). Generally, irrelevant signals would hinder the accurate investigation of the relationship between neural activity and movement behaviors. This raises concerns about whether irrelevant signals could conceal some critical facts about neural encoding and decoding mechanisms.

If the answer is yes, a natural question arises: what critical facts about neural encoding and decoding would irrelevant signals conceal? In terms of neural encoding, irrelevant signals may mask some small neural components, making their encoded information difficult to detect (*Moreno-Bote et al., 2014*), thereby misleading us to neglect the role of these signals, leading to a partial understanding of neural mechanisms. For example, at the single-neuron level, weakly tuned neurons are often assumed to contain little information and not analyzed (*Georgopoulos et al., 1986*; *Hochberg et al., 2012*; *Wodlinger et al., 2015*; *Inoue et al., 2018*); at the population level, neural signals composed of lower variance principal components (PCs) are typically treated as noise and discarded (*Churchland et al., 2012*; *Gallego et al., 2018*; *Gallego et al., 2020*; *Cunningham and Yu, 2014*). So, do these ignored signals truly contain little information, or do they appear that way only because they are obscured by irrelevant signals? And what's the role of these ignored signals? In terms of neural decoding, irrelevant signals would significantly complicate the information readout (*Pitkow et al., 2015*; *Yang et al., 2021*), potentially hindering the discovery of the true readout mechanism of behaviorally relevant responses. Specifically, in motor cortex, in what form (linear or nonlinear) downstream neurons readout behavioral information is an open question. Current studies typically use noisy raw signals for decoding behavioral information (*Georgopoulos et al., 1986*; *Hochberg et al., 2012*; *Wodlinger et al., 2015*; *Glaser et al., 2020*; *Willsey et al., 2022*). The linear readout is biologically plausible and widely used (*Georgopoulos et al., 1986*; *Hochberg et al., 2012*; *Wodlinger et al., 2015*), but recent studies (*Glaser et al., 2020*; *Willsey et al., 2022*) demonstrate nonlinear readout outperforms linear readout. So which readout scheme is the motor cortex more likely to adopt for decoding information from behaviorally relevant signals? Whether irrelevant signals are the culprits for the performance gap observed with raw signals? Unfortunately, all the above issues remain unclear.

One approach to address the above issues is to accurately separate behaviorally relevant and irrelevant signals at both single-neuron and single-trial levels and then analyze noise-free behaviorally relevant signals, which enables us to gain a more accurate and comprehensive understanding of the underlying neural mechanisms. However, this approach is hampered by the fact that the ground truth of behaviorally relevant signals is unknown, which makes the definition, extraction, and validation of behaviorally relevant signals a challenging task. As a result, methods of accurate separation remain elusive to date. Existing methods for extracting behaviorally relevant patterns at the single-trial level mainly focus on the latent population level (*Sani et al., 2021*; *Hurwitz, 2021*; *Zhou, 2020*) rather than the single-neuron level, and they extract neural activities based on assumptions about specific neural properties, such as linear or nonlinear dynamics (*Sani et al., 2021*; *Hurwitz, 2021*). Although these methods have shown promising results, they fail to capture other parts of behaviorally relevant neural activity that do not meet their assumptions, thereby providing an incomplete picture of behaviorally relevant neural activity. Some studies (*Kobak et al., 2016*; *Rouse and Schieber, 2018*) are able to extract behaviorally relevant neural signals at the single-neuron level, but they utilize trial-averaged responses, thereby losing the single-trial information. To overcome these limitations and obtain accurate behaviorally relevant signals at both single-neuron and single-trial levels, we propose a novel framework that defines, extracts, and validates behaviorally relevant signals by simultaneously considering such signals' encoding (behaviorally relevant signals should be similar to raw signals to preserve the underlying neuronal properties) and decoding (behaviorally relevant signals should contain behavioral information as much as possible) properties (see Methods and *Figure 1*). This framework establishes a prerequisite foundation for the subsequent detailed analysis of neural mechanisms.

Here, we conducted experiments using datasets recorded from the motor cortex of three monkeys performing different reaching tasks, where the behavioral variable is movement kinematics. After signal separation by our approach, we first explored how the presence of behaviorally irrelevant signals affects the analysis of neural activity. We found that behaviorally irrelevant signals account for a large amount of trial-to-trial neuronal variability, and are evenly distributed across the neural dimensions

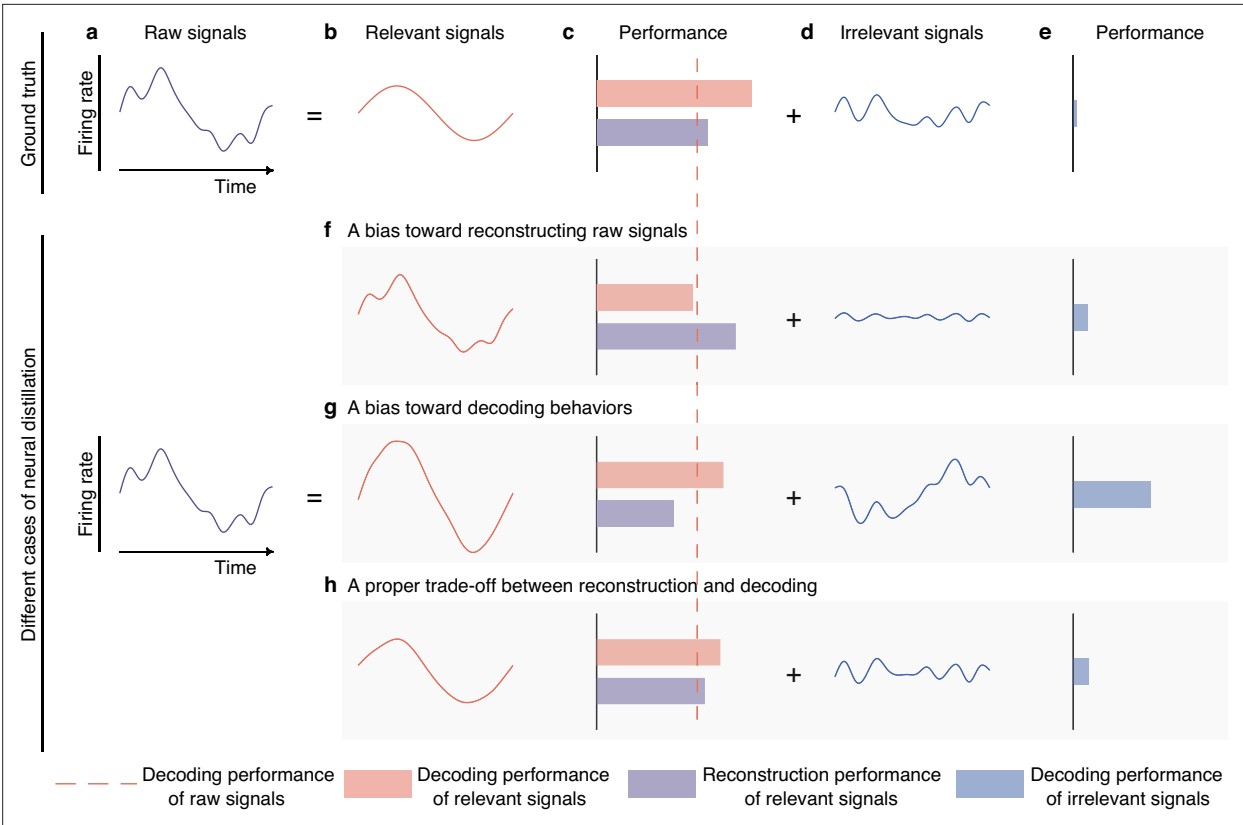

**Figure 1.** Semantic illustration of extracting and validating behaviorally relevant signals. (**a–e**) The ideal decomposition of raw signals. (**a**) The temporal neuronal activity of raw signals, where x-axis denotes time, and y-axis represents firing rate. Raw signals are decomposed to relevant (**b**) and irrelevant (**d**) signals. The red dotted line indicates the decoding performance of raw signals. The red and blue bars represent the decoding performance of relevant and irrelevant signals. The purple bar represents the reconstruction performance of relevant signals, which measures the neural similarity between generated signals and raw signals. The longer the bar, the larger the performance. The ground truth of relevant signals decodes information perfectly (**c**, red bar) and is similar to raw signals to some extent (**c**, purple bar), and the ground truth of irrelevant signals contains little behavioral information (**e**, blue bar). (**f–h**) Three different cases of behaviorally relevant signals distillation. (**f**) When the model is biased toward generating relevant signals that are similar to raw signals, it will achieve high reconstruction performance, but the decoding performance will suffer due to the inclusion of too many irrelevant signals. As it is difficult for models to extract complete relevant signals, the residuals will also contain some behavioral information. (**g**) When the model is biased toward generating signals that prioritize decoding over similarity to raw signals, it will achieve high decoding performance, but the reconstruction performance will be low. Meanwhile, the residuals will contain a significant amount of behavioral information. (**h**) When the model balances the trade-off of decoding and reconstruction capabilities of relevant signals, both decoding and reconstruction performance will be good, and the residuals will only contain a little behavioral information.

The online version of this article includes the following figure supplement(s) for figure 1:

**Figure supplement 1.** Semantic overview of distill-variational autoencoder (d-VAE).

**Figure supplement 2.** Visualization of latent variables.

of behaviorally relevant signals. Then, we explored whether irrelevant signals conceal some facts of neural encoding and decoding. For neural encoding, irrelevant signals obscure the behavioral information encoded by neural responses, especially for neural responses with a large degree of nonlinearity. Surprisingly, neural responses that are usually ignored (weakly tuned neurons and neural signals composed of small variance PCs) actually encode rich behavioral information in complex nonlinear ways. These responses underpin an unprecedented neuronal redundancy and reveal that movement behaviors are distributed in a higher-dimensional neural space than previously thought. In addition, we found that the integration of smaller and larger variance PCs results in a synergistic effect, allowing the smaller variance PC signals that cannot be linearly decoded to significantly enhance the linear decoding performance, particularly for finer speed control. This finding suggests that lower variance PC signals are involved in regulating precise motor control. For neural decoding, irrelevant signals complicate information readout. Strikingly, when uncovering small neural components obscured by

irrelevant signals, linear decoders can achieve comparable decoding performance with nonlinear decoders, providing strong evidence for the presence of linear readout in motor cortex. Together, our findings reveal unexpected complex encoding but simple decoding mechanisms in the motor cortex. Finally, our study also has implications for developing accurate and robust brain-machine interfaces (BMIs) and, more generally, provides a powerful framework for separating behaviorally relevant and irrelevant signals, which can be applied to other cortical data to uncover more neural mechanisms masked by behaviorally irrelevant signals.

## Results

### Framework for defining, extracting, and validating behaviorally relevant neural signals

#### What are behaviorally relevant neural signals?

Since the ground truth of behaviorally relevant signals is unknown, their precise definition is not yet well established. Before a definition is established, it is essential to first differentiate between relevant and irrelevant signals. Behaviorally irrelevant signals refer to those not directly associated with the behavioral variables of interest and may include noise or signals from variables of no interest. In contrast, behaviorally relevant signals refer to those directly related to the behavioral variables of interest.

Here, we define behaviorally relevant signals based on the following two requirements: (1) they should closely resemble raw signals to preserve the underlying neuronal properties, without becoming so similar that they include irrelevant signals (encoding requirement), and (2) they should contain behavioral information as much as possible (decoding requirement). Signals that meet both requirements are considered effective behaviorally relevant signals.

In this study, we assume raw signals (*Figure 1a*) are additively composed of behaviorally relevant (*Figure 1b*) and irrelevant (*Figure 1d*) signals. Thus, behaviorally irrelevant signals are derived by subtracting the behaviorally relevant signals from raw signals.

#### How to extract behaviorally relevant signals?

One way to extract behaviorally relevant signals is to use a distillation model to generate them from raw signals while considering the remaining signals as behaviorally irrelevant. However, due to the unknown ground truth of behaviorally relevant signals, a key challenge for the model is to determine the optimal degree of similarity between the generated signals and raw signals. If the generated signals are too similar to raw signals, they may contain a large amount of irrelevant information, which would hinder the exploration of neural mechanisms. Conversely, if the generated signals are too dissimilar to raw signals, they may lose behaviorally relevant information, also hindering the exploration of neural mechanisms. Therefore, finding the appropriate prior regularization knowledge to constrain the generated signals to resemble raw signals appropriately is key to modeling. We have formalized this extraction process as the following optimization problem:

$$\min_{\boldsymbol{x}_r} \mathrm{E}\left(\boldsymbol{x}_r, \boldsymbol{x}\right) + \mathrm{R}(\boldsymbol{x}_r), \tag{1}$$

where $\boldsymbol{x}$ denotes raw signals, $\boldsymbol{x}_r$ denotes generated signals, $\mathrm{E}(\cdot, \cdot)$ denotes reconstruction error, $\mathrm{R}(\cdot)$ denotes regularization loss. The regularization constraint on the generated signals $\mathrm{R}(\boldsymbol{x}_r)$ is crucial for accurately extracting behaviorally relevant signals. However, existing works (*Sani et al., 2021*; *Hurwitz, 2021*; *Zhou, 2020*) have not identified and addressed this key challenge.

To overcome this challenge, we exploited the trade-off between the similarity of generated signals to raw signals (encoding requirement) and their decoding performance of behaviors (decoding requirement) to extract effective behaviorally relevant signals (for details, see Methods and *Figure 1—figure supplement 1*). The core assumption of our model is that behaviorally irrelevant signals are noise relative to behaviorally relevant signals, and thereby irrelevant signals would degrade the decoding generalization of generated behaviorally relevant signals. Based on this assumption, we imposed decoding constraints to the generated signals $\boldsymbol{x}_r$ to minimize the inclusion of irrelevant signals, which is the operation used for modeling $\mathrm{R}(\boldsymbol{x}_r)$.

Generally, the distillation model is faced with three cases: a bias toward reconstructing raw signals (*Figure 1f*), a bias toward decoding behaviors (*Figure 1g*), and a proper trade-off between reconstruction and decoding (*Figure 1h*). If the distillation model is biased toward extracting signals similar to raw signals, the distilled behaviorally relevant signals will contain an excessive amount of behaviorally irrelevant information, affecting the decoding generalization of these signals (*Figure 1f*). If the model is biased toward extracting parsimonious signals that are discriminative for decoding, the distilled signals will not be similar enough to raw signals, and some redundant but useful signals will be left in the residuals (*Figure 1g*), making irrelevant signals contain much behavioral information. Using face recognition as an example, if a model can accurately identify an individual using only the person's eyes (assuming these are the most useful features), other useful information such as the nose or mouth will be left in the residuals, which could also be used to identify the individual. Neither of these two cases is desirable because the former loses decoding performance, while the latter loses some useful neural signals, which are not conducive to our subsequent analysis of the relationship between behaviorally relevant signals and behaviors. The behaviorally relevant signals we want should be similar to raw signals and preserve the behavioral information maximally, which can be obtained by balancing the encoding and decoding properties of generated behaviorally relevant signals (*Figure 1h*).

## How to validate behaviorally relevant signals?

To validate the effectiveness of the distilled signals, we proposed three criteria. The first criterion is that the decoding performance of the behaviorally relevant signals (red bar, *Figure 1*) should surpass that of raw signals (the red dotted line, *Figure 1*). Since decoding models, such as deep neural networks, are more prone to overfit noisy raw signals than behaviorally relevant signals, the distilled signals should demonstrate better decoding generalization than the raw signals. The second criterion is that the behaviorally irrelevant signals should contain minimal behavioral information (blue bar, *Figure 1*). This criterion can assess whether the distilled signals maximally preserve behavioral information from the opposite perspective and effectively exclude undesirable cases, such as over-generated and under-generated signals. Specifically, in the case of over-generation, suppose $z = x + y$, where $z$, $x$, and $y$ represent raw, relevant, and irrelevant signals, respectively. If the distilled relevant signals $\hat{x}$ are added extra signals $m$ which do not exist in the real behaviorally relevant signals, i.e., $\hat{x} = x + m$, then the corresponding residuals $\hat{y}$ will be equal to the ideal irrelevant signals $y$ plus the negative extra signals $-m$, namely, $\hat{y} = y - m$, thus the residuals $\hat{y}$ contain the amount of information preserved by negative extra signals $-m$. Similarly, in the case of under-generation, if the distilled behaviorally relevant signals are incomplete and lose some useful information, this lost information will also be reflected in the residuals. In these cases, the distilled signals are not suitable for analysis. The third criterion is that the distilled behaviorally relevant signals should be similar to raw signals to maintain essential neuronal properties (purple bar, *Figure 1*). If the distilled signals do not resemble raw signals, they fail to retain the fundamental characteristics of raw signals, which are not qualified for subsequent analysis. Overall, if the distilled signals satisfy the above three criteria, we consider the distilled signals to be effective.

## d-VAE extracts effective behaviorally relevant signals

To demonstrate the effectiveness of our model (distill-variational autoencoder [d-VAE]) in extracting behaviorally relevant signals, we conducted experiments on the synthetic dataset where the ground truth of relevant and irrelevant signals are already known (see Methods) and three benchmark datasets with different paradigms (*Figure 2a, e, and i*; see Methods for details), and compared d-VAE with four other distillation models, including pi-VAE (*Zhou, 2020*), PSID (*Sani et al., 2021*), TNDM (*Hurwitz, 2021*), and LFADS (*Pandarinath et al., 2018*). Specifically, we first applied these distillation models to raw signals to obtain the distilled behaviorally relevant signals, considering the residuals as behaviorally irrelevant signals. We then evaluated the decoding $R^2$ between the predicted velocity and actual velocity of the two partition signals using a linear Kalman filter (KF) and a nonlinear artificial neural network (ANN) and measured the neural similarity between behaviorally relevant and raw signals.

Overall, d-VAE successfully extracts effective behaviorally relevant signals that meet the three criteria outlined above on both synthetic (*Figure 2—figure supplement 1*) and real data (*Figure 2*). On the synthetic data (*Figure 2—figure supplement 1*), results show that d-VAE can strike an effective balance between the reconstruction and decoding performance of generated signals to extract

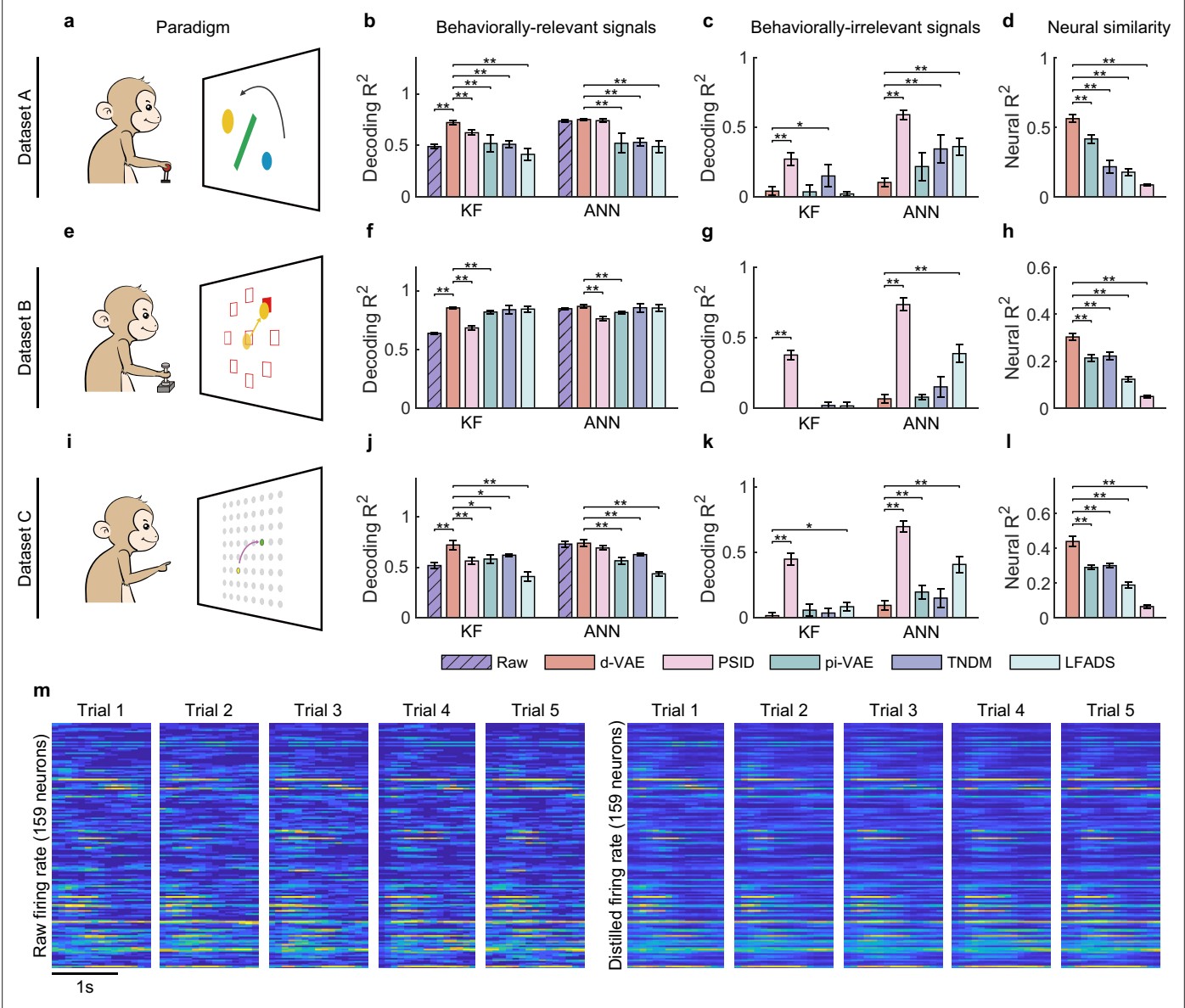

**Figure 2.** Evaluation of separated signals. (**a–d**) Results for dataset A. (**a**) The obstacle avoidance paradigm. (**b**) The decoding $R^2$ between true velocity and predicted velocity of raw signals (purple bars with slash lines) and behaviorally relevant signals obtained by distill-variational autoencoder (d-VAE) (red), PSID (pink), pi-VAE (green), TNDM (blue), and LFADS (light green). Error bars denote mean ± standard deviation (s.d.) across five cross-validation folds. Asterisks represent significance of Wilcoxon rank-sum test with *p<0.05, **p<0.01. (**c**) Same as (**b**), but for behaviorally irrelevant signals obtained by five different methods. (**d**) The neural similarity ($R^2$) between raw signals and behaviorally relevant signals extracted by d-VAE, PSID, pi-VAE, TNDM, and LFADS. Error bars represent mean ± s.d. across five cross-validation folds. Asterisks indicate significance of Wilcoxon rank-sum test with **p<0.01. (**e–h and i–l**). Same as (**a–d**), but for dataset B with the center-out paradigm (**e**) and dataset C with the self-paced reaching paradigm (**i**). (**m**) The firing rates of raw signals and distilled signals obtained by d-VAE in five held-out trials under the same condition of dataset B.

The online version of this article includes the following figure supplement(s) for figure 2:

**Figure supplement 1.** Evaluation of separated signals on the synthetic dataset.

effective relevant signals that are similar to the ground truth relevant signals, meanwhile removing effective irrelevant signals that resemble the ground truth irrelevant signals (*Figure 2—figure supplement 1a–g*), and outperforms other distillation models (*Figure 2—figure supplement 1h–k*). On the real data, specifically in the obstacle avoidance task (*Figure 2a*), the monkey is required to move the ball from the start point (blue) to the target point (yellow) without hitting the obstacle. For the decoding performance of behaviorally relevant signals (*Figure 2b*), the signals distilled by d-VAE

outperform the raw signals (purple bars with slash lines) and the signals distilled by all other distillation models (PSID, pink; pi-VAE, green; TNDM, blue; and LFADS, light green) with the KF as well as the ANN. For the decoding performance of behaviorally irrelevant signals (*Figure 2c*), behaviorally irrelevant signals obtained by d-VAE achieves the lowest decoding performance compared with behaviorally irrelevant signals obtained by other approaches. Therefore, the combination of dVAE's highest decoding performance for behaviorally relevant signals and lowest decoding performance for behaviorally irrelevant signals demonstrate its superior ability to extract behaviorally relevant signals from noisy signals. For the neural similarity between behaviorally relevant and raw signals (*Figure 2d*), the distilled signals obtained by d-VAE achieve the highest performance among competitors (p<0.01, Wilcoxon rank-sum test). Similar results were obtained for the center-out task (*Figure 2e–h*) and the self-paced reaching task (*Figure 2i–l*), indicating the consistency of d-VAE's distillation ability across a range of motor tasks. To provide a more intuitive illustration of the similarity between raw and distilled signals, we displayed the firing rate of neuronal activity in five trials under the same condition (*Figure 2m*), and results clearly show that the firing pattern of distilled signals is similar to the corresponding raw signals.

In summary, d-VAE distills effective behaviorally relevant signals that preserve behavioral information maximally and are similar to raw signals. Meanwhile, the behaviorally irrelevant signals discarded by d-VAE contain a little behavioral information. Therefore, these signals are reliable for exploring the encoding and decoding mechanisms of relevant signals.

## How do behaviorally irrelevant signals affect the analysis of neural activity at the single-neuron level?

Following signal separation, we first explored how behaviorally irrelevant signals affect the analysis of neural activity at the single-neuron level. Specifically, we examined the effect of irrelevant signals on two critical properties of neuronal activity: the preferred direction (PD) (*Georgopoulos et al., 1986*) and trial-to-trial variability. Our objective was to know how irrelevant signals affect the PD of neurons and whether irrelevant signals contribute significantly to neuronal variability.

To explore how irrelevant signals affect the PD of neurons, we first calculated the PD of both raw and distilled signals separately and then quantified the PD deviation by the angle difference between these two signals. Results show that the PD deviation increases as the neuronal $R^2$ decreases (red curve, *Figure 3a and e* and *Figure 3—figure supplement 1a*). It is worth noting that when using $R^2$ to describe neurons, it indicates the extent to which neuronal activity is explained by the linear encoding model (*Collinger et al., 2013*; *Wodlinger et al., 2015*). Neurons with larger $R^2$ (strongly linear-tuned neurons) exhibit stable PDs with signal distillation (see example PDs in the inset), while neurons with smaller $R^2$ (weakly linear-tuned neurons) show a larger PD deviation. These results indicate that irrelevant signals have a small effect on strongly tuned neurons but a large effect on weakly tuned neurons. One possible reason for the larger PD deviation in weakly tuned neurons is that they have a lower degree of linear encoding but a higher degree of nonlinear encoding, and highly nonlinear structures are more susceptible to interference from irrelevant signals (*Nogueira et al., 2023*). Moreover, after filtering out the behaviorally irrelevant signals, the cosine tuning fit ($R^2$) of neurons increases (p<10$^{-20}$, Wilcoxon signed-rank test; *Figure 3b and f* and *Figure 3—figure supplement 1b*), indicating that irrelevant signals reduce the neurons' tuning expression. Notably, even after removing the interference of irrelevant signals, the $R^2$ of neurons remains relatively low and varies among neurons. These results demonstrate that the linear encoding model only explains a small fraction of neural responses, and neuronal activity encodes behavioral information in complex nonlinear ways.

To investigate whether irrelevant signals significantly contribute to neuronal variability, we compared the neuronal variability (measured with the Fano factor [FF]; *Churchland et al., 2010*) of relevant and raw signals. Results show that the condition-averaged FF of each neuron of distilled signals is lower than that of raw signals (p<10$^{-20}$, Wilcoxon signed-rank test; *Figure 3c and g*), and the mean (broken line) and median FFs of all neurons under different conditions are also significantly lower than those of raw signals (p<0.01, Wilcoxon signed-rank test; *Figure 3d and h*), indicating that irrelevant signals significantly contribute to neuronal variability. We then visualized the single-trial neuronal activity of example neurons under different conditions (*Figure 3i* and *Figure 3—figure supplement 2*). Results demonstrate that the patterns of relevant signals are more consistent and stable across different trials than raw signals, and the firing activity of irrelevant signals varies randomly. These results indicate that

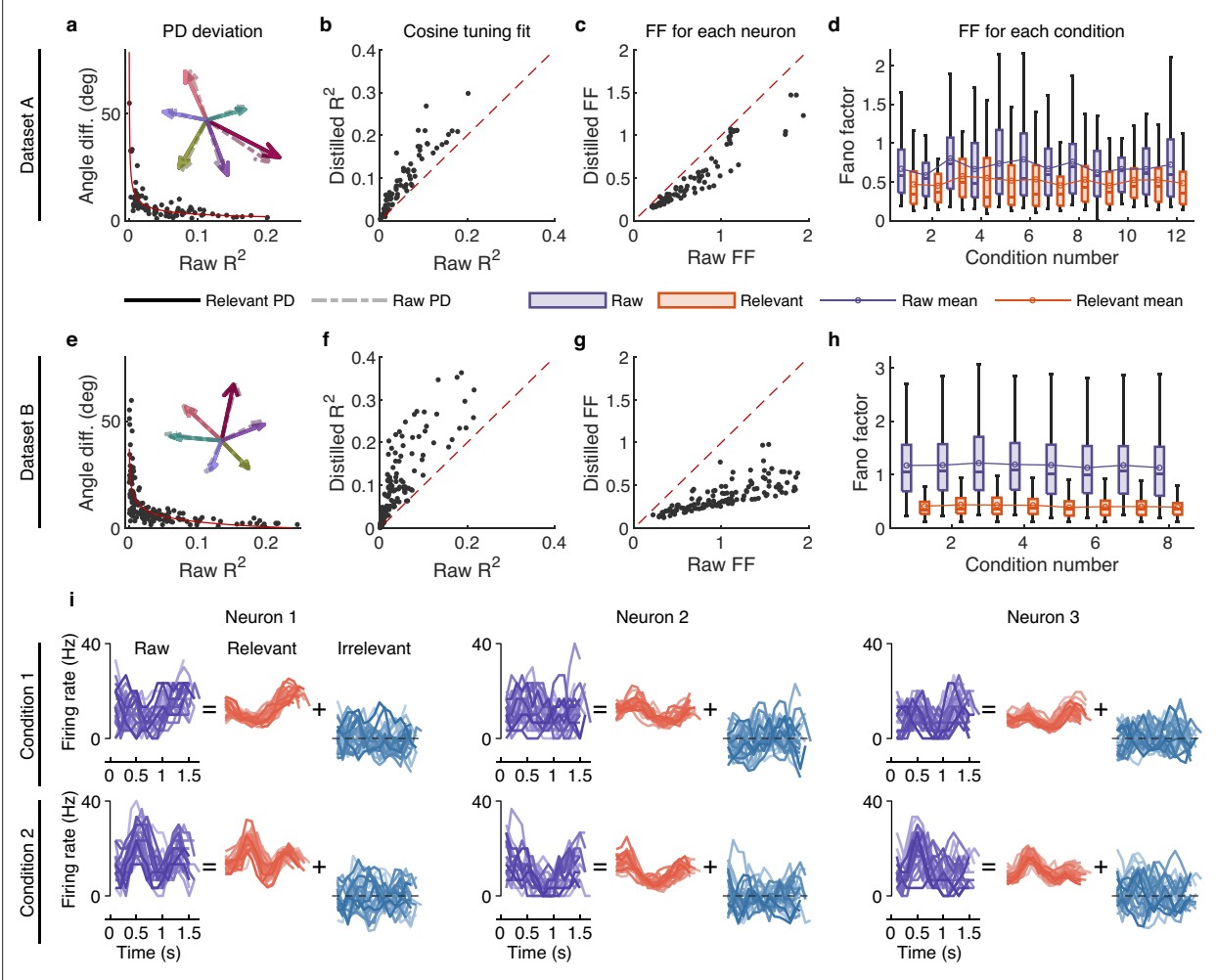

**Figure 3.** The effect of irrelevant signals on analyzing neural activity at the single-neuron level. (**a–d**) Results for dataset A. (**a**) The angle difference (AD) of preferred direction (PD) between raw and distilled signals as a function of the $R^2$ of raw signals. When employing $R^2$ to characterize neurons, it indicates the extent to which neuronal activity is explained by the linear encoding model. Smaller $R^2$ neurons have a lower capacity for linearly tuning (encoding) behaviors, while larger $R^2$ neurons have a higher capacity for linearly tuning (encoding) behaviors. Each black point represents a neuron (n=90). The red curve is the fitting curve between $R^2$ and AD. Five example larger $R^2$ neurons' PDs are shown in the inset plot, where the solid and dotted line arrows represent the PDs of relevant and raw signals, respectively. (**b**) Comparison of the cosine tuning fit ($R^2$) before and after distillation of single neurons (black points), where the x-axis and y-axis represent neurons' $R^2$ of raw and distilled signals, respectively. (**c**) Comparison of neurons' Fano factor (FF) averaged across conditions of raw (x-axis) and distilled (y-axis) signals, where FF is used to measure the neuronal variability of different trials in the same condition. (**d**) Boxplots of raw (purple) and distilled (red) signals under different conditions for all neurons (12 conditions). Boxplots represent medians (lines), quartiles (boxes), and whiskers extending to ±1.5 times the interquartile range. The broken lines represent the mean FF across all neurons. (**e–h**) Same as (**a–d**), but for dataset B (n=159, 8 conditions). (**i**) Example of three neurons' raw firing activity decomposed into behaviorally relevant and irrelevant parts using all trials under two conditions (2 of 8 directions) in held-out test sets of dataset B.

The online version of this article includes the following figure supplement(s) for figure 3:

**Figure supplement 1.** The effect of irrelevant signals on relevant signals at the single-neuron level.

**Figure supplement 2.** The firing activity of example neurons.

irrelevant signals significantly contribute to neuronal variability, and eliminating the interference of irrelevant signals enables us to observe the changes in neural pattern more accurately.

## How do behaviorally irrelevant signals affect the analysis of neural activity at the population level?

The neural population structure is an essential characteristic of neural activity. Here, we examined how behaviorally irrelevant signals affect the analysis of neural activity at the population level, including

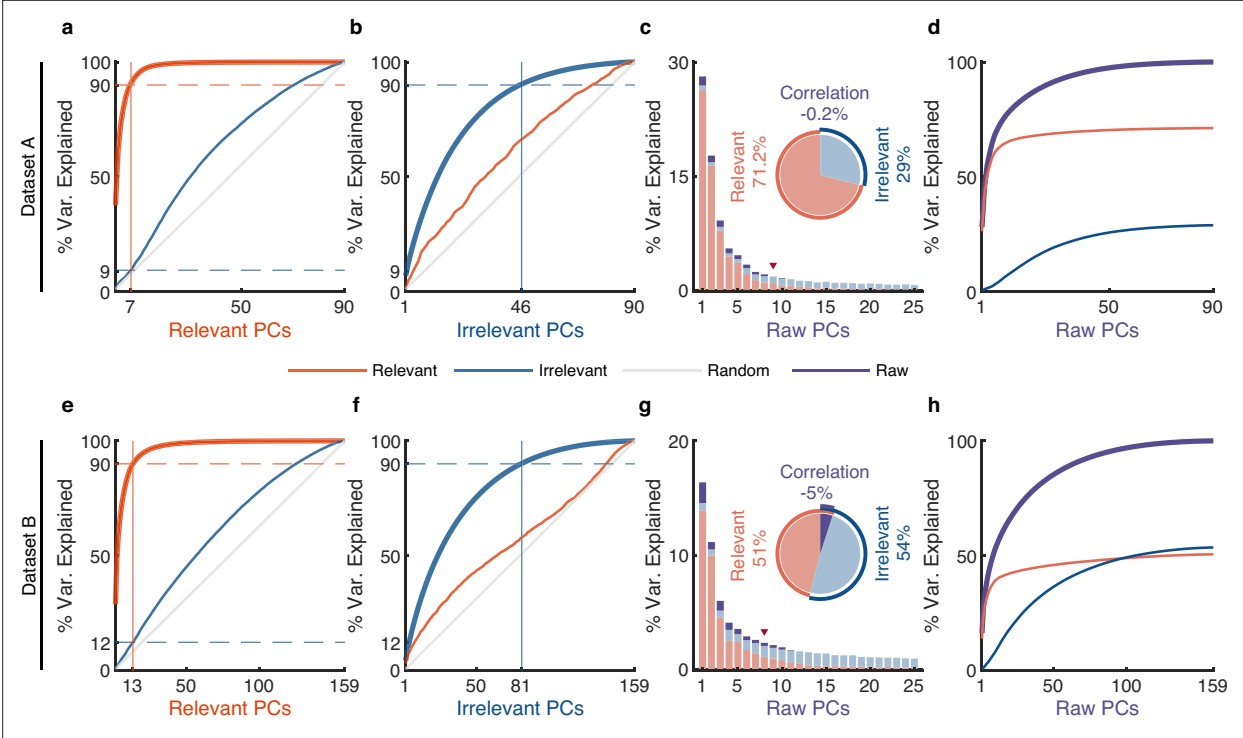

**Figure 4.** The effect of irrelevant signals on analyzing neural activity at the population level. (**a–d**) Results for dataset A. (**a**) The cumulative variance curve for different signals, including relevant signals (red), irrelevant signals (blue), and random Gaussian noise $\mathcal{N}(\mathbf{0}, \mathbf{I})$ (gray, representing the chance level), projected onto the principal components (PCs) of relevant signals. Specifically, principal component analysis (PCA) is applied to relevant signals to get relevant PCs. Subsequently, the three types of signals are projected onto these relevant PCs to obtain their respective cumulative variance curves. The thick lines represent the cumulative variance explained for the signals on which PCA has been performed, while the thin lines represent the variance explained by those PCs for other signals. The horizontal dotted lines represent the percentage of variance explained. The vertical lines indicate the number of dimensions that accounted for 90% of the variance in behaviorally relevant (left) and irrelevant (right) signals. For convenience, we defined the PC subspace describing the top 90% variance as the primary subspace and the subspace capturing the last 10% variance as the secondary subspace. (**b**) Same as (**a**), but for irrelevant PCs. (**c**) The composition of raw signals and each raw PC. Specifically, PCA is applied to the raw signals to obtain raw PCs. Then, the relevant and irrelevant signals are projected onto these raw PCs to determine the variance of the raw signals explained by each type of signal. The bar plot shows the composition of each raw PC. The inset pie plot shows the overall proportion of raw signals, where red, blue, and purple colors indicate relevant signals, irrelevant signals, and the correlation between relevant and relevant signals. The PC marked with a red triangle indicates the last PC where the variance of relevant signals is greater than or equal to that of irrelevant signals. (**d**) The cumulative variance explained by raw PCs for different signals, where the thick line represents the cumulative variance explained for raw signals (purple), while the thin line represents the variance explained for relevant (red) and irrelevant (blue) signals. (**e–h**) Same as (**a–d**), but for dataset B.

The online version of this article includes the following figure supplement(s) for figure 4:

**Figure supplement 1.** The effect of irrelevant signals on analyzing neural activity at the population level.

**Figure supplement 2.** The effect of irrelevant signals obtained by pi-VAE on analyzing neural activity at the population level.

**Figure supplement 3.** The rotational dynamics of raw, relevant, and irrelevant signals.

**Figure supplement 4.** The cumulative variance curve for raw and behaviorally relevant signals.

four aspects: (1) the population properties of relevant and irrelevant signals, (2) the subspace overlap relationship between the two signal components, (3) how the two partitions contribute to raw signals, and (4) the difference in population properties between raw and distilled signals.

To explore the population properties of relevant and irrelevant signals, we separately applied principal component analysis (PCA) on each partition to obtain the corresponding cumulative variance curve in a descending variance order. Our results show that the primary subspace (capturing the top 90% variance) of relevant signals (thick red line, *Figure 4a and e* and *Figure 4—figure supplement 1a*) is only explained by a few dimensions (7, 13, and 9 for each dataset), indicating that the primary part of behaviorally relevant signals exists in a low-dimensional subspace. In contrast, the primary subspace of irrelevant signals (thick blue line, *Figure 4b and f* and *Figure 4—figure supplement*

*1b*) requires more dimensions (46, 81, and 59). The variance distribution of behaviorally irrelevant signals across dimensions (thick blue line, *Figure 4b and f* and *Figure 4—figure supplement 1b*) is more even than behaviorally relevant signals (thick red line, *Figure 4a and e* and *Figure 4—figure supplement 1a*) but not as uniform as Gaussian noise $\mathcal{N}(\mathbf{0}, \mathbf{I})$ (thin gray line, *Figure 4b and f* and *Figure 4—figure supplement 1a*), indicating that irrelevant signals are not pure noise but rather bear some significant structure, which may represent information from other irrelevant tasks.

To investigate the subspace overlap between relevant and irrelevant signals, we calculated how many variances of irrelevant signals can be captured by relevant PCs by projecting irrelevant signals onto relevant PCs and vice versa (*Elsayed et al., 2016*; *Rouse and Schieber, 2018*; *Jiang et al., 2020*) (see Methods). We found that the variance of irrelevant signals increases relatively uniformly over relevant PCs (thin blue line, *Figure 4a and e* and *Figure 4—figure supplement 1a*), like random noise's variance accumulation explained by relevant PCs (thin gray line, *Figure 4a and e* and *Figure 4—figure supplement 1a*); similar results are observed for relevant signals explained by irrelevant PCs (thin red line, *Figure 4b and f* and *Figure 4—figure supplement 1b*). These results indicate that relevant PCs cannot match the informative dimensions of irrelevant signals and vice versa, suggesting the dimensions of behaviorally relevant and irrelevant signals are unrelated. It is worth mentioning that the signals obtained by pi-VAE are in contrast to our findings. Its results show that a few relevant PCs can explain a considerable variance of irrelevant signals (thin red line, *Figure 4—figure supplement 2b, f, j*), which indicates that the relevant and irrelevant PCs are closely related. The possible reason is that the pi-VAE leaves many relevant signals within the irrelevant signals. Notably, *Figure 4a and e* and *Figure 4—figure supplement 1a* show that the behaviorally relevant primary subspace captures only a minor portion of the variance from irrelevant signals when they are projected onto it (9%, 12%, and 9%), indicating that the primary subspace of behaviorally relevant signals is nearly orthogonal to irrelevant space.

To investigate the composition of raw signals by the two partitions, we performed PCA on raw neural signals to obtain raw PCs, and then projected the relevant and irrelevant signals onto these PCs to assess the proportion of variance of raw signals explained by each partition. First, we analyzed the overall proportion of relevant and irrelevant signals that constitute the raw signals (the inset pie plot, *Figure 4c and g* and *Figure 4—figure supplement 1c*). The variance of the raw signals is composed of three parts: the variance of relevant signals, the variance of irrelevant signals, and the correlation between relevant and irrelevant signals (see Methods). The results demonstrate that the irrelevant signals account for a large proportion of raw signals, suggesting the motor cortex encodes considerable information that is not related to the measured behaviors. In addition, there is only a weak correlation between relevant and irrelevant signals, implying that behaviorally relevant and irrelevant signals are nearly uncorrelated.

We then examined the proportions of relevant and irrelevant signals in each PC of raw signals. We found that relevant signals (red) occupy the dominant proportions in the larger variance raw PCs (before the PC marked with a red triangle), while irrelevant signals (blue) occupy the dominant proportions in the smaller variance raw PCs (after the PC marked with a red triangle) (*Figure 4c and g* and *Figure 4—figure supplement 1c*). Similar results are observed in the accumulation of each raw PC (*Figure 4d and h* and *Figure 4—figure supplement 1d*). Specifically, the results show that the variance accumulation of raw signals (purple line) in larger variance PCs is mainly contributed by relevant signals (red line), while irrelevant signals (blue line) primarily contribute to the lower variance PCs. These results demonstrate that irrelevant signals have a small effect on larger variance raw PCs but a large effect on smaller variance raw PCs. This finding eliminates the concern that irrelevant signals would significantly affect the top few PCs of raw signals and thus produce inaccurate conclusions. To further validate this finding, we used the top six PCs as jPCA (*Churchland et al., 2012*) did to examine the rotational dynamics of distilled and raw signals (*Figure 4—figure supplement 3*). Results show that the rotational dynamics of distilled signals are similar to those of raw signals.

Finally, to directly compare the population properties of raw and relevant signals, we plotted the cumulative variance curves of raw and relevant signals (*Figure 4—figure supplement 4*). Results (upper left corner curves, *Figure 4—figure supplement 4*) show that the cumulative variance curve of relevant signals (red line) accumulates faster than that of raw signals (purple line) in the preceding larger variance PCs, indicating that the variance of the relevant signal is more concentrated in the larger variance PCs than that of raw signals. Furthermore, we found that the dimensionality of primary

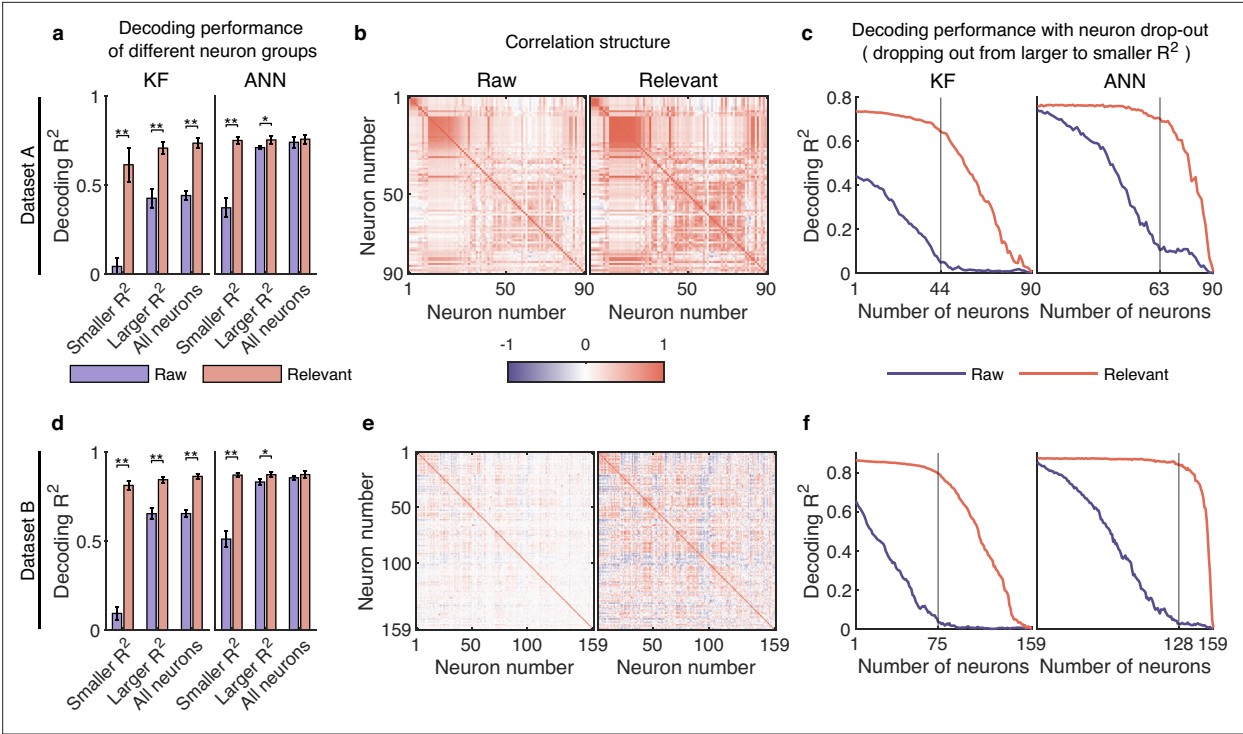

**Figure 5.** Smaller $R^2$ neurons encode rich behavioral information in complex nonlinear ways. (**a–c**) Results for dataset A. (**a**) The comparison of decoding performance between raw (purple) and distilled signals (red) with different neuron groups, including smaller $R^2$ neuron ($R^2 <= 0.03$), larger $R^2$ neuron ($R^2 > 0.03$), and all neurons. Error bars indicate mean ± standard deviation (s.d.) across five cross-validation folds. Asterisks denote significance of Wilcoxon rank-sum test with \*p<0.01, \*\*p<0.01. (**b**) The correlation matrix of all neurons of raw (left) and behaviorally relevant (right) signals. Neurons are ordered to highlight correlation structure (details in Methods). (**c**) The decoding performance of Kalman filter (KF) (left) and artificial neural network (ANN) (right) with neurons dropped out from larger to smaller $R^2$. The vertical gray line indicates the number of dropped neurons at which raw and behaviorally relevant signals have the greatest performance difference. (**d–f**) Same as (**a–c**), but for dataset B.

The online version of this article includes the following figure supplement(s) for figure 5:

**Figure supplement 1.** Neural responses usually considered to contain little information actually encode rich behavioral information in complex nonlinear ways.

**Figure supplement 2.** Using synthetic data to demonstrate that conclusions are not a by-product of distill-variational autoencoder (d-VAE).

subspace of raw signals (26, 64, and 45 for datasets A, B, and C) is significantly higher than that of behaviorally relevant signals (7, 13, and 9), indicating that using raw signals to estimate the neural dimensionality associated with behaviors leads to an overestimation.

## Distilled behaviorally relevant signals uncover that smaller $R^2$ neurons encode rich behavioral information in complex nonlinear ways

The results presented above regarding PDs (*Figure 3* and *Figure 3—figure supplement 1*) demonstrate that irrelevant signals significantly impact smaller $R^2$ neurons and weakly impact larger $R^2$ neurons. Under the interference of irrelevant signals, it is difficult to explore the amount of behavioral information in neuronal activity. Given that we have accurately separated the behaviorally relevant and irrelevant signals, we explored whether irrelevant signals would mask some encoded information of neuronal activity, especially for smaller $R^2$ neurons.

To answer the question, we divided the neurons into two groups of smaller $R^2$ ($R^2 <= 0.03$) and larger $R^2$ ($R^2 > 0.03$), and then used decoding models to assess how much information is encoded in raw and distilled signals. As shown in *Figure 5a*, for the smaller $R^2$ neuron group, both KF and ANN decode behavioral information poorly on raw signals, but achieve high decoding performance using relevant signals. Specifically, the KF decoder (left plot, *Figure 5a*) improves the decoding $R^2$ significantly from 0.044 to 0.616 (improves by about 1300%, Wilcoxon rank-sum test) after signal distillation; the ANN decoder (right plot, *Figure 5a*) improves from 0.374 to 0.753 (improves by about

100%, Wilcoxon rank-sum test). For the larger $R^2$ neuron group, the decoding performance of relevant signals with ANN does not improve much compared with the decoding performance of raw signals, but the decoding performance of relevant signals with KF is significantly better than that of raw signals (p<0.01, Wilcoxon rank-sum test). Similar results are obtained with datasets B (*Figure 5d*) and C (*Figure 5—figure supplement 1a*). These results indicate that irrelevant signals mask behavioral information encoded by neuronal populations, especially for smaller $R^2$ neurons with a higher degree of nonlinearity, and that smaller $R^2$ neurons actually encode rich behavioral information.

The fact that the smaller $R^2$ neurons encode rich information seems unexpected, and interestingly, we cannot obtain this rich information solely by distilling smaller $R^2$ neurons. This observation gives rise to two alternative scenarios. The first is that larger $R^2$ neurons introduce additional signals to smaller $R^2$ neurons, which they do not inherently possess, resulting in an excessive amount of behavioral information within the smaller $R^2$ neurons. The second is that the smaller $R^2$ neurons inherently possess a substantial amount of information, and larger $R^2$ neurons utilize their neural activity, which is correlated with that of small $R^2$ neurons, to aid in restoring the small $R^2$ neurons' original appearance; this process is analogous to image denoising, where damaged noisy pixels necessitate the assistance of their correlated, clean neighboring pixels to recover their original appearance. We initially tested the first scenario and found it to be unsupported for two key reasons. First, our model enforces that distilled neuronal activity closely resembles the corresponding original neuronal activity, effectively preventing the generation of arbitrarily shaped neuronal activity, such as that of other neurons. As shown in *Figure 3i* and *Figure 3—figure supplement 2*, our distilled relevant neuronal activity exhibits a high degree of similarity to the corresponding raw neuronal activity. To assess whether the distilled neurons exhibit the highest similarity to the corresponding raw neurons, we compared the neural similarity ($R^2$) of each distilled neuron to all raw neurons. The results indicate that 78/90 (87%, dataset A), 153/159 (96%, dataset B), and 91/91 (100%, dataset C) distilled neurons are most similar to the corresponding neurons. The remaining distilled neurons rank among the top four in similarity to the corresponding neurons, further confirming the close resemblance of distilled neuronal activity to the corresponding raw neuronal activity. Second, as we emphasized in the section on validating behaviorally relevant signals with the second criterion, if this large amount of information is compensated by other neurons, the residuals should also contain a large amount of information. However, as illustrated in *Figure 2c, g, and k*, the residuals contain only little information. Therefore, based on these two reasons, the first scenario is rejected. Then, we tested the second scenario. To verify this scenario, we conducted experiments using synthetic data with known ground truth (see Methods). In this dataset, small $R^2$ neurons inherently contained a substantial amount of information but were obscured by noise, making them undecodable. We aimed to assess whether d-VAE could recover the lost information and restore the damaged neuronal activity. The results demonstrate that, with the assistance of large $R^2$ neurons, d-VAE effectively recovers a significant amount of information that is obscured by noise (*Figure 5—figure supplement 2a*). Additionally, the distilled signals exhibit a remarkable improvement in neural similarity to the ground truth signals compared to the raw signals (p<0.01, Wilcoxon rank-sum test; *Figure 5—figure supplement 2b*). Therefore, these results support the second scenario and collectively confirm that smaller $R^2$ neurons indeed contain rich behavioral information, and this finding is not a by-product of d-VAE.

Given that both smaller and larger $R^2$ neurons encode rich behavioral information, it is worth noting that the sum of the decoding performance of smaller $R^2$ neurons and larger $R^2$ neurons is significantly greater than that of all neurons for relevant signals (red bar, *Figure 5a and d* and *Figure 5—figure supplement 1a*), demonstrating that movement parameters are encoded very redundantly in neuronal population. In contrast, we cannot find this degree of neural redundancy in raw signals (purple bar, *Figure 5a and d* and *Figure 5—figure supplement 1a*) because the encoded information of smaller $R^2$ neurons are masked by irrelevant signals. Therefore, these smaller $R^2$ neurons, which are usually ignored, are actually useful and play a critical role in supporting neural redundancy. Generally, cortical redundancy can arise from neuronal correlations, which are critical for revealing certain aspects of neural circuit organization (*Yatsenko et al., 2015*). Accordingly, we visualized the ordered correlation matrix of neurons (see Methods) for both raw and relevant signals (*Figure 5b and e* and *Figure 5—figure supplement 1b*) and found that the neuronal correlation of relevant signals is stronger than that of raw signals. These results demonstrate that irrelevant signals weaken the neuronal correlation, which may hinder the accurate investigation of neural circuit organization.

Considering the rich redundancy and strong correlation of neuronal activity, we wondered whether the neuronal population could utilize redundant information from other neurons to exhibit robustness under the perturbation of neuronal destruction. To investigate this question, we evaluated the decoding performance of dropping out neurons from larger $R^2$ to smaller $R^2$ on raw and relevant signals. The results (*Figure 5c and f* and *Figure 5—figure supplement 1c*) show that the decoding performance of the KF and ANN on raw signals (purple line) decreases steadily before the number of neurons marked (vertical gray line), and the remaining smaller $R^2$ neurons decode behavioral information poorly. In contrast, even if many neurons are lost, the decoding performance of KF and ANN on relevant signals (red line) maintains high accuracy. This finding indicates that behaviorally relevant signals are robust to the disturbance of neuron drop-out, and smaller $R^2$ neurons play a critical role in compensating for the failure of larger $R^2$ neurons. In contrast, this robustness cannot be observed in raw signals because irrelevant signals mask neurons' information and weaken their correlation. Notably, the ANN outperforms the KF when only smaller $R^2$ neurons are left (*Figure 5c and f* and *Figure 5—figure supplement 1c*), suggesting that smaller $R^2$ neurons can fully exploit their nonlinear ability to cope with large-scale neuronal destruction.

## Distilled behaviorally relevant signals uncover that signals composed of smaller variance PCs encode rich behavioral information in complex nonlinear ways

The results presented above regarding subspace overlap (*Figure 4* and *Figure 3—figure supplement 1*) show that irrelevant signals have a small impact on larger variance PCs but dominate smaller variance PCs. Therefore, we aimed to investigate whether irrelevant signals would mask some encoded information of neural population, especially signals composed of smaller variance PCs.

To answer the question, we compared the decoding performance of raw and distilled signals with different raw PC groups. Specifically, we first divided the raw PCs into two groups, i.e., smaller variance PCs and larger variance PCs, defined by ratio of relevant to irrelevant signals in the raw PCs (the red triangle, see *Figure 4c and g* and *Figure 3—figure supplement 1c*). Then, we projected raw and distilled signals onto these two PC groups and got the corresponding signals. Results show that, for the smaller variance PC group, both KF and ANN achieve much better performance on distilled signals than raw signals (p<0.01, Wilcoxon rank-sum test, for ANN), whereas for the larger variance PC group, the decoding performance of relevant signals does not improve a lot compared with the decoding performance of raw signals (see *Figure 6a and d* and *Figure 5—figure supplement 1d*). These results demonstrate that irrelevant signals mask the behavioral information encoded by different PC groups, especially for signals composed of smaller variance PCs (smaller variance PC signals), and smaller variance PC signals actually encode rich behavioral information.

The above results are based on raw PCs. However, raw PCs are biased by irrelevant signals and thus cannot faithfully reflect the characteristics of relevant signals. As we have successfully separated the behaviorally relevant signals, we aimed to explore how behavioral information of distilled signals is distributed across relevant PCs. To do so, we used decoding models to evaluate the amount of behavioral information contained in cumulative PCs of relevant signals (using raw signals as a comparison). The cumulative variance explained by PCs in descending and ascending order of variance and the dimensionality corresponding to the top 90% variance signals (called primary signals) and the last 10% variance signals (called secondary signals) are shown in *Figure 4—figure supplement 4*.

Here, we first investigated secondary signals' decoding ability solely by accumulating PCs from smaller to larger variance. The results show that, for relevant signals, KF can hardly decode behavioral information solely using secondary signals (red line; left plot, *Figure 6b and e* and *Figure 5—figure supplement 1e*), but ANN can decode rich information (red line; right plot, *Figure 6b and e* and *Figure 5—figure supplement 1e*). These results indicate that smaller variance PC signals encode rich information in complex nonlinear ways. In contrast, when using raw signals composed of the same number of dimensions as the secondary signals (purple line, *Figure 6b and e* and *Figure 5—figure supplement 1e*), the amount of information identified by ANN is significantly smaller than that of relevant secondary signals (p<0.01, Wilcoxon rank-sum test). These results demonstrate that signals composed of these neural dimensions actually encode rich behavioral information, and irrelevant signals make them seem insignificant, indicating that behavioral information is distributed in a higher-dimensional subspace than expected from raw signals.

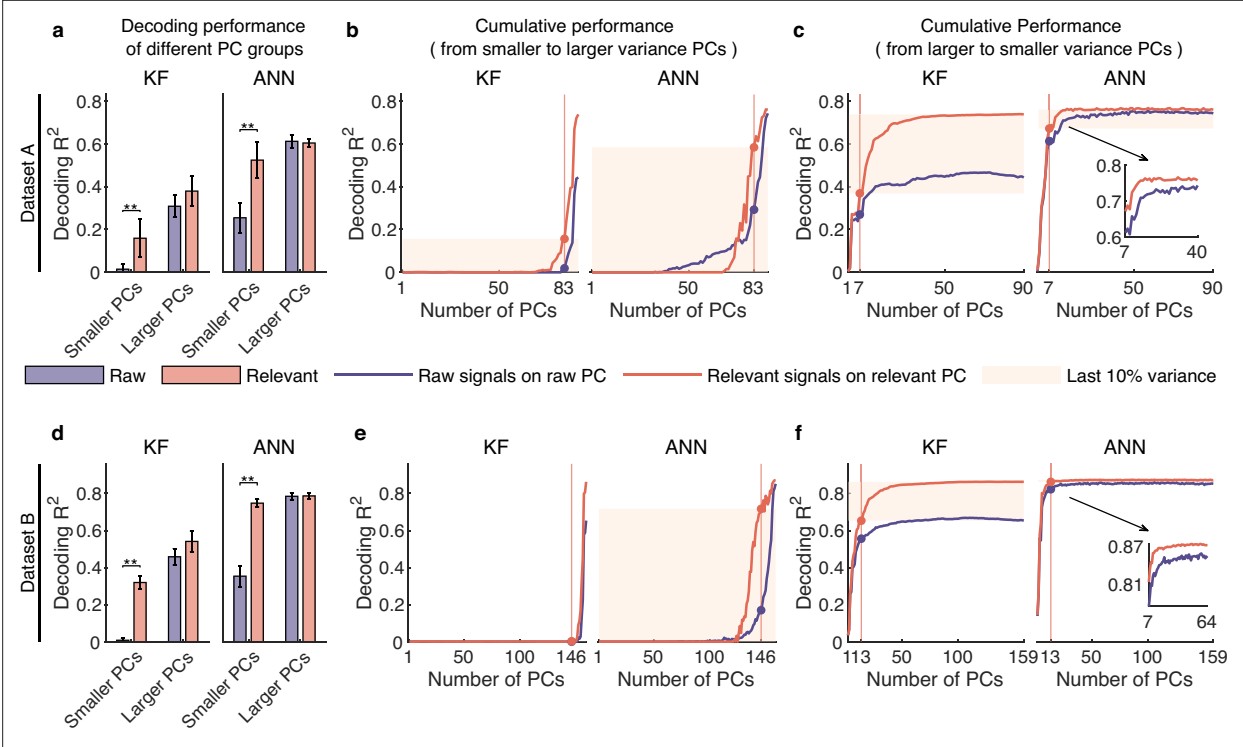

**Figure 6.** Signals composed of smaller variance principal components (PCs) encode rich behavioral information in complex nonlinear ways. (**a–c**) Results for dataset A. (**a**) The comparison of decoding performance between raw (purple) and distilled signals (red) composed of different raw PC groups, including smaller variance PCs (the proportion of irrelevant signals that make up raw PCs is higher than that of relevant signals), larger variance PCs (the proportion of irrelevant signals is lower than that of relevant ones). Error bars indicate mean ± standard deviation (s.d.) across five cross-validation folds. Asterisks denote significance of Wilcoxon rank-sum test with *p<0.01, **p<0.01. (**b**) The cumulative decoding performance of signals composed of cumulative PCs that are ordered from smaller to larger variance using Kalman filter (KF) (left) and artificial neural network (ANN) (right). The red patches indicate the decoding ability of the last 10% variance of relevant signals. (**c**) The cumulative decoding performance of signals composed of cumulative PCs that are ordered from larger to smaller variance using KF (left) and ANN (right). The red patches indicate the decoding gain of the last 10% variance signals of relevant signals superimposing on their top 90% variance signals. The inset shows the partially enlarged plot for view clearly. (**d–f**) Same as (**a–c**), but for dataset B.

The online version of this article includes the following figure supplement(s) for figure 6:

**Figure supplement 1.** Smaller variance principal component (PC) signals preferentially improve lower-speed velocity.

We then investigated the effect of superimposing secondary signals on primary signals by accumulating PCs from larger to lower variance. The results (***Figure 6c and f*** and ***Figure 5—figure supplement 1f***) show that secondary signals improve the decoding performance of ANN a little but improve the decoding performance of KF a lot. The discrepancy between the two decoders reflects their different abilities to utilize the information within the signal. KF cannot use the nonlinear information in primary signals as effectively as ANN can and thus require secondary signals to improve decoding performance. Notably, KF shows steady growth in decoding performance on relevant signals across 10–30 dimensions, and requires approximately 30–40 dimensions to achieve performance saturation. These results demonstrate that these smaller variance PC signals actually encode behavioral information, and suggest that behavioral information exists in a higher-dimensional subspace than anticipated from raw signals. Interestingly, we can find that although secondary signals nonlinearly encode behavioral information and are decoded poorly by linear decoders, they considerably improve KF performance by superimposing on primary signals (left plot, ***Figure 6c and f*** and ***Figure 5—figure supplement 1f***); and the sum of the sole decoding performance of primary and secondary signals is lower than the decoding performance of full signals. These results indicate that the combination of smaller and larger variance PCs produces a synergy effect (***Narayanan et al., 2005***), enabling secondary signals that cannot be linearly decoded to improve the linear decoding performance.

Finally, considering the substantial enhancement in KF decoding performance when superimposing the secondary signals on the primary ones, we explored which aspect of movement parameters was most improved. In BMIs, directional control has achieved great success (*Georgopoulos et al., 1986*; *Hochberg et al., 2012*), but precise speed control, especially at lower speeds such as hold or stop, has always been challenging (*Wodlinger et al., 2015*; *Inoue et al., 2018*). Thus, we hypothesized that these signals might improve the lower-speed velocity. To test this, we divided samples into lower-speed and higher-speed regions and assessed which region improved the most by superimposing the secondary signals (see details in Methods). After superimposing the secondary signals, the absolute improvement ratio of the lower-speed region is significantly higher than that of the higher-speed region (p<0.05, Wilcoxon rank-sum test; *Figure 6—figure supplement 1a, b, and c*). Furthermore, we visualized the relative improvement ratio of five example trials for the two regions, and the results (*Figure 6—figure supplement 1d*) demonstrate that secondary signals significantly improve the estimation of lower speed. These results demonstrate that the secondary signals enhance the lower-speed control, suggesting that smaller variance PC signals may be involved in regulating precise motor control.

## Distilled behaviorally relevant signals potentially suggest that motor cortex may use a linear readout mechanism to generate movement behaviors

Understanding the readout mechanism of the motor cortex is crucial for both neuroscience and neural engineering, which remains unclear. By filtering out the interference of behaviorally irrelevant signals, we found a stunning result: the linear decoder KF achieves comparable performance to that of the nonlinear decoder ANN (p=0.10, 0.15, and 0.55 for datasets A, B, and C, Wilcoxon rank-sum test; *Figure 2b, f, and j*). Considering the decoding performance and model complexity (the simplicity principle, also called Occam's razor), movement behaviors are more likely to be generated by the linear readout, suggesting linear readout may be performed in the motor cortex.

Given the significant improvement in linear decoding performance, one might doubt that it is our distillation model that makes signals that are inherently nonlinearly decodable become linearly decodable. In practice, this situation does not hold for two reasons. First, our criterion that irrelevant signals should contain minimal information can effectively exclude this situation. Specifically, if this situation occurs, the model would significantly modify the structure of the generated signals, causing a deviation from the structure of the ground truth signals. Consequently, these uncharacterized or modified ground truth signals would remain within the residuals, resulting in residuals that contain a substantial amount of information. To illustrate this, consider an example where $z = x + y = n^2 + y$, with $z$, $x$, $y$, and $n$ representing raw signals, relevant signals, irrelevant signals, and behavioral variables, respectively. If the distilled relevant signals are $\hat{x} = n$, the corresponding irrelevant signals are $n^2 - n + z$. Clearly, the distilled signal can be linearly decoded, but this results in the residuals containing a large amount of information. However, as demonstrated in *Figure 2c, g, and k*, the irrelevant signals obtained by d-VAE only contain little information, thus excluding this situation. Second, our synthetic experiments offer additional evidence supporting the conclusion that d-VAE does not make inherently nonlinearly decodable signals become linearly decodable ones. As depicted in *Figure 5—figure supplement 2c*, there exists a significant performance gap between KF and ANN when decoding the ground truth signals of smaller $R^2$ neurons (p<0.01, Wilcoxon rank-sum test). KF exhibits notably low performance, leaving substantial room for compensation by d-VAE. However, following processing by d-VAE, KF's performance of distilled signals fails to surpass its already low ground truth performance and remains significantly inferior to ANN's performance (p<0.01, Wilcoxon rank-sum test). These results collectively confirm that our approach does not convert signals that are inherently nonlinearly decodable into linearly decodable ones.

In summary, these findings demonstrate that behaviorally irrelevant signals significantly complicate the readout of behavioral information and provide compelling evidence supporting the notion that the motor cortex may use a linear readout mechanism to generate movement behaviors.

## Discussion

In this study, we proposed a new perspective for studying neural mechanisms, namely, using separated accurate behaviorally relevant signals instead of raw signals; and we provided a novel distillation framework to define, extract, and validate behaviorally relevant signals. By separating behaviorally relevant and irrelevant signals, we found that neural responses previously considered to contain little information actually encode rich behavioral information in complex nonlinear ways, and they play an important role in neural encoding and decoding. Furthermore, we found that linear decoders can achieve comparable performance to that of nonlinear decoders, providing compelling evidence for the presence of linear readout in the motor cortex. Overall, our results reveal unexpected complex encoding but simple decoding mechanisms in the motor cortex.

### Signal separation by d-VAE

Behaviorally relevant patterns can be extracted either at single-neuron or latent neural population levels. In our study, we focused on the single-neuron level, aiming to preserve the underlying properties of individual neurons. By maintaining the properties of each neuron, researchers can investigate how the neuronal population performs when one of the neurons is destroyed. This kind of analysis is particularly useful in closed-loop stimulation experiments that use electrophysiological (*Sun et al., 2022*) or optogenetic (*Zhang et al., 2023*) interventions. Furthermore, behaviorally relevant signals also allow for population-level analysis and provide clean benchmark signals to test and compare the variance capture ability of different hypothesis-driven models.

At the single-neuron level, it is common practice to use trial-averaged neuronal responses of the same task parameters to analyze neural mechanisms (*Kobak et al., 2016*; *Rouse and Schieber, 2018*). However, trial averaging sacrifices single-trial information, thereby providing an incomplete characterization of neural activity. Furthermore, trial-averaged responses still contain a significant amount of behaviorally irrelevant signals caused by uninstructed movements (*Musall et al., 2019*), which can lead to a contaminated version of behaviorally relevant signals. In contrast, our model is capable of extracting clean behaviorally relevant neural activity for every single trial. At the latent population level, existing latent variable models (*Sani et al., 2021*; *Pandarinath et al., 2018*; *Yu, 2008*; *Zhou, 2020*; *Hurwitz, 2021*) focus on modeling some specific properties of latent population representations, such as linear or nonlinear dynamics (*Sani et al., 2021*; *Pandarinath et al., 2018*; *Churchland et al., 2012*; *Hurwitz, 2021*), temporal smoothness (*Yu, 2008*), and interpretability (*Zhou, 2020*). Since these models make restrictive assumptions involving characterizing specific neural properties, they fail to capture other parts of behaviorally relevant signals that do not meet their assumptions, providing no guarantee that the generated signals preserve behavioral information maximally. In contrast, our objective is to extract accurate behaviorally relevant signals that closely approximate the ground truth relevant signals as much as possible. To ensure this, we deliberately impose constraints on the model, ensuring that it generates signals that retain neuronal properties while preserving behavioral information to the highest degree possible. Notably, the pivotal operation of striking a balance between the reconstruction and decoding performance of generated signals to extract relevant signals is a distinctive feature absent in other models. At the population level, dimensionality reduction methods aided by task parameters (*Kobak et al., 2016*; *Schneider et al., 2023*) are another important way to discover the latent neural embeddings relevant to task parameters, which may provide new insight into neural representations. In contrast with this class of methods, our model focuses on the signal level, not the latent embedding level.

Although we made every effort, our model is still not able to perfectly extract behaviorally relevant neural signals, resulting in a small amount of behavioral information leakage in the residuals. Nevertheless, the signals distilled by our model are reliable, and the minor imperfections do not affect the conclusions drawn from our analysis. In the future, better models can be developed to extract behaviorally relevant signals more accurately, such as incorporating multiple time step information (*Pandarinath et al., 2018*; *Sani et al., 2021*; *Hurwitz, 2021*) and contrastive learning (*Schneider et al., 2023*) or metric learning (*Li et al., 2021*) techniques into models.

## Implications for analyzing neural activity by separation

Studying neural mechanisms through noisy signals is akin to looking at flowers in a fog, which makes it difficult to discern the truth. Thus, removing the interference of irrelevant signals is necessary and beneficial for analyzing neural activity, whether at the single-neuron level or population level.

At the single-neuron level, trial-to-trial neuronal variability poses a significant challenge to identifying the actual neuronal pattern changes. The variability can arise from various sources, including meaningless noise (*Faisal et al., 2008*), meaningful but behaviorally irrelevant neural processes (*Musall et al., 2019*), and intrinsic components of neural encoding (*Walker et al., 2020*). However, it is still unclear to what extent each source contributes to the variability (*Faisal et al., 2008*). By separating behaviorally relevant and irrelevant parts, we could roughly determine the extent to which these two parts contribute to the variability and explore which type of variability these two parts may contain. Our results demonstrate that behaviorally irrelevant signals are a significant contributor to variability, which may include both meaningless noise and meaningful but behaviorally irrelevant signals as behaviorally irrelevant signals are not pure noise and may carry some structures (thick blue line, *Figure 4b and f* and *Figure 4—figure supplement 1b*). Notably, behaviorally relevant signals also exhibit some variability, which may arise from intrinsic components of neural encoding and provide the neural basis for motor learning (*Dhawale et al., 2017*). Moreover, eliminating the variability caused by irrelevant signals enables us to better observe and compare actual neuronal pattern changes and may facilitate the study of learning mechanisms (*Sadtler et al., 2014*; *Hennig et al., 2021*).

At the population level, the dimensionality of neural manifolds quantifies the degrees of freedom required to describe population activity without significant information loss (*Lee and Verleysen, 2007*; *Altan et al., 2021*). However, determining the dimensionality of neural manifolds associated with specific behaviors from raw signals is challenging since it is difficult to discern how many variances correspond to irrelevant signals, which often depend heavily on signal quality. A previous study (*Altan et al., 2021*) demonstrated, through simulation experiments involving different levels of noise, that such noise makes methods overestimate the neural dimensionality. Our results, consistent with theirs, indicate that using raw signals which include many irrelevant signals will cause an overestimation of the neural dimensionality (*Figure 4—figure supplement 4*). These findings highlight the need to filter out irrelevant signals when estimating the neural dimensionality. Furthermore, this perspective of signal separation has broader implications for other studies. For instance, researchers can isolate neural signals corresponding to different behaviors and explore their shared and exclusive patterns to uncover underlying common and unique mechanisms of different behaviors (*Gallego et al., 2018*).

## Implications for exploring neural mechanisms by separation

At the single-neuron level, previous studies (*Carmena et al., 2005*; *Narayanan et al., 2005*) have shown that neuronal ensembles redundantly encode movement behaviors in the motor cortex. However, our results reveal a significantly higher level of redundancy than previously reported. Specifically, prior studies found that the decoding performance steadily declines as neurons drop out, which is consistent with our results drawn from raw signals. In contrast, our results show that decoders maintain high performance on distilled signals even when many neurons drop out. Our findings reinforce the idea that movement behavior is redundantly encoded in the motor cortex and demonstrate that the brain is robust enough to tolerate large-scale neuronal destruction while maintaining brain function (*Alstott et al., 2009*).

At the population level, previous studies have proposed that motor control is achieved through low-dimensional neural manifolds, with analyses typically using between 6 and 15 PCs (*Churchland et al., 2012*; *Kaufman et al., 2014*; *Elsayed et al., 2016*; *Sadtler et al., 2014*; *Golub et al., 2018*; *Gallego et al., 2017*; *Gallego et al., 2020*). However, our results challenge this idea by showing that signals composed of smaller variance PCs nonlinearly encode a significant amount of behavioral information, and the number of useful PCs ranges from 30 to 40, far exceeding the usual number analyzed. These results suggest that behavioral information is distributed in a higher-dimensional neural space than previously thought. Interestingly, although smaller variance PC signals nonlinearly encode behavioral information, their behavioral information can be linearly decoded by superimposing them onto larger variance PC signals. This result is consistent with the finding that nonlinear mixed selectivity can yield high-dimensional neural responses and thus allow linear readout of behavioral information by downstream neurons (*Rigotti et al., 2013*; *Fusi et al., 2016*). Moreover, we found that smaller

variance PC signals can improve precise motor control, such as lower-speed control. Analogously, recent studies have found that smaller variance PCs of hand postures are task-dependent and relate to the precise and complex postures (*Yan et al., 2020*). These findings suggest that neural signals composed of lower variance PCs may be involved in the regulation of precise motor control.

In the motor cortex, in what form downstream neurons read out behavioral information is still an open question. Previous studies have shown that nonlinear readout is superior to linear readout on raw signals (*Naufel et al., 2019*; *Glaser et al., 2020*; *Willsey et al., 2022*). However, by filtering out the interference of behaviorally irrelevant signals, our study found that accurate decoding performance can be achieved through linear readout, suggesting that the motor cortex may perform linear readout to generate movement behaviors. Similar observations involving raw signals have been reported across various cortices, including the inferotemporal cortex (*Majaj et al., 2015*), perirhinal cortex (*Pagan et al., 2013*), and somatosensory cortex (*Nogueira et al., 2023*). These observations support the hypothesis that linear readout might serve as a general principle in the brain. However, further experiments are needed to verify this hypothesis across a wider range of cortical regions. In motor cortex, different neurons encode behavioral information with varying degrees of nonlinearity, exhibiting complex and heterogeneous response patterns. Despite this complexity of neural encoding, these responses allow for a linear readout of behavioral information. This phenomenon suggests that the complexity of encoding mechanisms may underlie the simplicity of decoding mechanisms.

About studying decoding mechanisms, recent studies (*Pitkow et al., 2015*; *Ganmor et al., 2015*; *Yang et al., 2021*) have focused on investigating how the brain decodes task information in the presence of noise. Unlike previous works, our research specifically explores the decoding mechanisms of behaviorally relevant signals rather than raw signals. We assume that the brain filters out irrelevant signals before decoding the relevant ones. This leads to the question of whether the brain actually adopts this strategy to access relevant signals. Given the existence of behaviorally relevant signals, it is reasonable to assume that the brain has intrinsic mechanisms to differentiate between relevant and irrelevant signals. There is growing evidence suggesting that the brain utilizes various mechanisms, such as attention and specialized filtering, to suppress irrelevant signals and enhance relevant signals (*Sreenivasan and Fiete, 2011*; *Schneider et al., 2018*; *Nakajima et al., 2019*). Therefore, it is plausible that the brain filters before decoding, thereby effectively accessing behaviorally relevant signals. Furthermore, our study reveals that irrelevant signals are the most critical factor affecting accurate and robust decoding, and achieving accurate and robust linear decoding requires weak neural responses. These findings have two important implications for developing accurate and robust BMIs: designing preprocessing filtering algorithms or developing decoding algorithms that include filtering out behaviorally irrelevant signals, and paying attention to the role of weak neural responses in motor control. More generally, our study provides a powerful framework for separating behaviorally relevant and irrelevant signals, which can be applied to other cortical data to uncover more hidden neural mechanisms.

## Methods
### Dataset and preprocessing
Three datasets with different paradigms are employed, including obstacle avoidance task dataset (*Wang et al., 2017*), center-out reaching task dataset (*Dyer et al., 2017*), and self-paced reaching task dataset (*O'Doherty, 2017*).

The first dataset (dataset A) is the obstacle avoidance dataset. An adult male Rhesus monkey was trained to use the joystick to move the computer cursor to bypass the obstacle and reach the target. Neural data were recorded from the monkey's upper limb area of the dorsal premotor (PMd) using a 96-electrode Utah array (Blackrock Microsystems Inc, USA). Multi-unit activity (MUA) signals are used in the present study. The corresponding behavioral data (velocity) were simultaneously collected. There are 2 days of data (20140106 and 20140107), and each day contains 171 trials on average. All animal handling procedures were authorized by the Animal Care Committee at Zhejiang University, China, and conducted following the Guide for Care and Use of Laboratory Animals (China Ministry of Health).

The second dataset (dataset B) is publicly available and provided by Kording Lab (*Dyer et al., 2017*). The monkey was trained to complete two-dimensional eight-direction center-out reaching

tasks. We used 2 days of data from subject C (20161007 and 20161011). Each day contains 190 trials on average. Neural data are spike-sorted PMd signals. The behavioral data were simultaneously collected in instantaneous velocity.

The third dataset (dataset C) is publicly available and provided by Sabes Lab (Zenodo dataset) (*O'Doherty, 2017*). An adult male Rhesus monkey was trained to finish self-paced reaching tasks within an 8-by-8 square grid. There are no inter-trial intervals during the experiment. Neural data were recorded from the monkey's primary motor cortex (M1) area with a 96-channel silicon microelectrode array. The neural data are the MUA signals. Hand velocity was obtained from the position through a discrete derivative. The recording period for the data (20170124 01) is about 10 min.

For all datasets, the neural signals were binned by a 100 ms sliding window without overlap. As a preprocess, we smoothed the neural signals using a moving average filter with three bins. We excluded some electrode activities with low mean firing rates (<0.5 Hz mean firing rates across all bins) and did not perform any other pre-selection to select neurons. For the computation of the FF, we chose 12 and 14 points as the thresholds of trial length for datasets A and B, respectively; trials with a length less than the threshold were discarded (discard about 7% and 2% trials for datasets A and B), trials longer than the threshold were truncated to threshold length from the starting point. Since dataset C has no trial information, FF is not calculated for this dataset. For the analysis of datasets A and B, we selected 1 day of these two datasets for analysis (20140107 for dataset A and 20161011 for dataset B).

## The synthetic dataset

The synthetic dataset is used to demonstrate that d-VAE can extract effective behaviorally relevant signals that are similar to the ground truth signals. The specific process of generating synthetic data is as follows. First, we randomly selected nine larger $R^2$ neurons from neurons that $R^2$ is greater than 0.1, and three smaller $R^2$ neurons from neurons that $R^2$ is lower than 0.01 of dataset B (20161011). Second, we used deep neural networks to learn the encoding model between movement kinematics (movement velocity of dataset B) and neural signals using onefold train data. The details of the networks are demonstrated as follows. The networks use two hidden layer multilayer perceptron (MLP) with 500 and 500 hidden units. The activation function is ReLU. A SoftPlus activation function follows the last layer of the networks. The reconstruction loss is the Poisson likelihood function. After learning the encoding model, we used the learned encoding model to generate the ground truth of behaviorally relevant signals from all kinematics data of dataset B. Then, we added white Gaussian noise to the behaviorally relevant signals such that the noisy signals have a signal-to-noise ratio of 7 dB. After adding noise, the $R^2$ of the three smaller $R^2$ neurons is lower than 0.03. We regarded the noisy signals as raw signals and the added Gaussian noise as behaviorally irrelevant signals. Finally, we separated the synthetic data into five folds for cross-validation model evaluation.

## Distill-VAE

Notation: $x \in \mathbb{R}^n$ denotes raw neural signals. $x_r \in \mathbb{R}^n$ represents behaviorally relevant signals. $x_i = x - x_r \in \mathbb{R}^n$ represents behaviorally irrelevant signals. $z \in \mathbb{R}^d$ denotes the latent neural representations. $z_{prior} \in \mathbb{R}^d$ denotes the prior latent neural representations. $y \in \mathbb{R}^k$ represents kinematics. $f : \mathbb{R}^n \to \mathbb{R}^d$ represents the inference model (encoder) of VAE. $g : \mathbb{R}^d \to \mathbb{R}^n$ represents the generative model (decoder) of VAE. $m : \mathbb{R}^k \to \mathbb{R}^d$ represents the mapping from kinematics to prior latent representations. $h : \mathbb{R}^d \to \mathbb{R}^k$ represents an affine mapping from latent representations to kinematics.

d-VAE is a generative model based on VAEs (*Kingma, 2013*), specially designed to extract behaviorally relevant signals from raw signals. The generative model of d-VAE is

$$p_\theta(x|y) = \int_z p_\theta(x, z|y)\mathrm{d}z = \int_z p_m(z|y)p_g(x|z)\mathrm{d}z, \qquad (2)$$

where $p_m(z|y)$ denotes the conditional prior distribution of latent variables given the kinematics parameterized by feedforward neural networks $m$, $p_g(x|z)$ denotes the conditional prior distribution of raw signals given the latent variables parameterized by feedforward neural networks $g$, $p_\theta(x, z|y)$ represents the joint distribution of raw signals and latent variables given the kinematics parameterized by parameters $\theta = (m, g)$, and $p_\theta(x|y)$ is the marginal distribution of raw signals parameterized by parameters $\theta$.

To learn the model, we need to maximize the evidence lower bound (ELBO) of $p_\theta(x|y)$:

$$\mathcal{L}_{ELBO}(\boldsymbol{x}) = \mathbb{E}_{z \sim q_\phi(z|\boldsymbol{x}, \boldsymbol{y})} \left[ p_\theta(\boldsymbol{x}|z) \right] - D_{KL}(q_\phi(z|\boldsymbol{x}, \boldsymbol{y})|p(z|\boldsymbol{y})) \leq \log p_\theta(\boldsymbol{x}|\boldsymbol{y}), \tag{3}$$

where the first term on the right-hand side of **Equation 3** is called reconstruction term, the second term is called regularization term, $q_\phi(z|\boldsymbol{x}, \boldsymbol{y})$ denotes inference model parameterized by parameters $\phi$, and $D_{KL}(\cdot|\cdot)$ denotes the Kullback-Leibler (KL) divergence. In d-VAE, we set $q_\phi(z|\boldsymbol{x}, \boldsymbol{y}) = q_f(z|\boldsymbol{x})$, where $f$ is the inference model parameterized by feedforward neural networks. Because during the test stage, we cannot obtain the ground truth of kinematics, and we need to use only raw signals to extract relevant signals. Note that d-VAE aims to extract behaviorally relevant signals from raw signals, not generate signals that are too similar to raw signals. Therefore, we modified the objective loss function based on $\mathcal{L}_{ELBO}(\boldsymbol{x})$ (see **Equation 10**).

To distill behaviorally relevant neural signals, d-VAE utilizes the trade-off between the decoding and reconstruction abilities of generated behaviorally relevant signals $\boldsymbol{x}_r$. The basic assumption is generated behaviorally relevant signals that contain behaviorally irrelevant signals harms their decoding ability. Our approach for distilling behaviorally relevant signals consists of three steps, including identifying latent representations $z$, generating behaviorally relevant signals $\boldsymbol{x}_r$, and decoding behaviorally relevant signals $\boldsymbol{x}_r$.

## Identifying latent representations

Identifying latent representations containing behaviorally relevant information is the crucial part because latent representations influence the subsequent generation. Effective representations are more likely to generate proper behaviorally relevant neural signals $\boldsymbol{x}_r$. d-VAE identifies latent representations with inference model $f$, i.e., μ, where μ and $\boldsymbol{\sigma}^2$ denote the mean and variance of latent representations; thus the posterior distribution is $q_f(z|\boldsymbol{x}) = \mathcal{N}(z|\boldsymbol{\mu}, \boldsymbol{\sigma}^2)$. Then, we guide latent representations containing behavioral information through an affine map $h : \mathbb{R}^d \to \mathbb{R}^k$ under the loss $\mathcal{L}_{dec1}$,

$$\mathcal{L}_{dec1} = \text{MSE}(h(\boldsymbol{\mu}), \boldsymbol{y}), \tag{4}$$

where $\text{MSE}(\cdot, \cdot)$ denotes mean squared loss. In other words, we encourage latent representations to decode kinematics to distill behaviorally relevant information. Here, we sample from the approximation posterior $\boldsymbol{x}_r = g(z)$ using $z \sim \boldsymbol{\mu} + \boldsymbol{\sigma} \odot \boldsymbol{\epsilon}$, where $\boldsymbol{\epsilon} \sim \mathcal{N}(\boldsymbol{0}, \boldsymbol{I})$ and $\odot$ denotes element-wise product. This sampling strategy is known as the reparameterization trick.

## Generating behaviorally relevant signals

After sampling latent representations $z$, we send latent representations to the generative model $g$ to generate behaviorally relevant neural signals $\boldsymbol{x}_r$, i.e., $\boldsymbol{x}_r = g(z)$. We use following loss to make behaviorally relevant signals reconstruct raw signals as much as possible:

$$\mathcal{L}_{rec} = \text{Poisson}(\boldsymbol{x}_r, \boldsymbol{x}), \tag{5}$$

where $\text{Poisson}(\cdot, \cdot)$ denotes Poisson negative log likelihood loss. It is important to note that optimizing the generation of behaviorally relevant signals to accurately reconstruct noisy raw signals may result in the inclusion of many behaviorally irrelevant signals in the generated signals, which deviates from our initial goal of extracting behaviorally relevant signals. In the following subsection, we will introduce how to avoid generating behaviorally irrelevant signals.

## Decoding behaviorally relevant signals

As mentioned above, if the generation of behaviorally relevant signals $\boldsymbol{x}_r$ is only guided by $\mathcal{L}_{rec}$, generated signals may contain more behaviorally irrelevant signals. To avoid generated signals containing behaviorally irrelevant signals, we introduce decoding loss $\mathcal{L}_{dec2}$ to constrain $\boldsymbol{x}_r$ to decode behavioral information. The basic assumption is that behaviorally irrelevant signals act like noise for decoding behavioral information and are detrimental to decoding. Thus, there is a trade-off between neural reconstruction and decoding ability of $\boldsymbol{x}_r$: the more behaviorally irrelevant signals $\boldsymbol{x}_r$ contains, the more decoding performance $\boldsymbol{x}_r$ loses. Then, we send the $\boldsymbol{x}_r$ to the encoder $f$ and obtain the mean and variance of latent representations, i.e., $[\boldsymbol{\mu}_r; \boldsymbol{\sigma}_r^2] = f(\boldsymbol{x}_r)$. The decoding loss $\mathcal{L}_{dec2}$ is as follows:

$$\mathcal{L}_{dec2} = \text{MSE}(h(\boldsymbol{\mu}_r), \boldsymbol{y}). \tag{6}$$

We use the same networks $f$ and $h$ for $\boldsymbol{x}_r$ and $\boldsymbol{x}$ in our experiment, because $\boldsymbol{x}_r$ can act as data augmentation and make $f$ distill robust representations without increasing model parameters. In addition, we combine the two decoding loss as one loss:

$$\mathcal{L}_{dec} = \frac{1}{2}(\mathcal{L}_{dec1} + \mathcal{L}_{dec2}). \tag{7}$$

## Learning the prior distribution with behavioral information

The prior distribution of latent representation is crucial because inappropriate prior assumptions can degrade latent representations and generated neural signals. Vanilla VAE uses a Gaussian prior $\mathcal{N}(\boldsymbol{0}, \boldsymbol{I})$ to regularize the space of latent representation $\boldsymbol{z}$. However, in neuroscience, the distribution of latent representations is unknown and may exceed the scope of Gaussian. Therefore, we adopt neural networks $m$ to learn the prior distribution with kinematics $\boldsymbol{y}$, i.e., $[\boldsymbol{\mu}_{prior}; \boldsymbol{\sigma}_{prior}^2] = m(\boldsymbol{y})$ and thus $p_m(\boldsymbol{z}|\boldsymbol{y}) = \mathcal{N}(z_{prior}|\boldsymbol{\mu}_{prior}, \boldsymbol{\sigma}_{prior}^2)$. The prior distribution $p_m(\boldsymbol{z}|\boldsymbol{y})$ and approximation posterior distribution $q_f(\boldsymbol{z}|\boldsymbol{x})$ are aligned by the KL divergence:

$$\mathcal{L}_{KL} = D_{KL}(p_m(\boldsymbol{z}|\boldsymbol{y})|q_f(\boldsymbol{z}|\boldsymbol{x})) = D_{KL}(\mathcal{N}(\boldsymbol{z}|\boldsymbol{\mu}, \boldsymbol{\sigma}^2)|\mathcal{N}(z_{prior}|\boldsymbol{\mu}_{prior}, \boldsymbol{\sigma}_{prior}^2)). \tag{8}$$

In this case, the distribution of $p(\boldsymbol{z})$ is

$$p(\boldsymbol{z}) = \int \hat{p}(\boldsymbol{y}) p_m(\boldsymbol{z}|\boldsymbol{y}) \mathrm{d}\boldsymbol{y} = \sum_{j=1}^{N} (\frac{1}{N} \sum_{i=1}^{N} \delta(\boldsymbol{y}^{(j)} - \boldsymbol{y}^{(i)})) p_m(\boldsymbol{z}|\boldsymbol{y}^{(j)}), \tag{9}$$

where $\hat{p}(\boldsymbol{y}) = \frac{1}{N} \sum_{i=1}^{N} \delta(\boldsymbol{y} - \boldsymbol{y}^{(i)})$ represents the empirical distribution of behavioral variables $\boldsymbol{y}$, $N$ denotes the number of samples, $\delta(\cdot)$ denotes Dirac delta function. Thus, $p(\boldsymbol{z})$ corresponds to a Gaussian mixture model with $N$ components, which is theoretically a universal approximator of continuous probability densities. Since $z_{prior}$ and $\boldsymbol{z}$ are aligned and the generative network $g$ models the relationship between $\boldsymbol{z}$ and $\boldsymbol{x}_r$, this is equivalent to indirectly establishing a neural encoding model from $\boldsymbol{y}$ to $\boldsymbol{x}_r$. Thus, we can observe the change of $z_{prior}$ and $\boldsymbol{x}_r$ by changing $\boldsymbol{y}$ and can better understand the encoding mechanism of neural signals.

## End-to-end optimization

d-VAE is optimized in an end-to-end manner under the following loss:

$$\mathcal{L} = \mathcal{L}_{rec} + \beta \mathcal{L}_{KL} + \alpha \mathcal{L}_{dec}, \tag{10}$$

where $\beta$ and $\alpha$ are hyperparameters, $\beta$ is used to adjust the weight of KL divergence, and $\alpha$ determines the trade-off between reconstruction loss $\mathcal{L}_{rec}$ and decoding loss $\mathcal{L}_{dec}$. Given that the ground truth of latent variable distribution is unknown, even a learned prior distribution might not accurately reflect the true distribution. We found the pronounced impact of the KL divergence would prove detrimental to the decoding and reconstruction performance. As a result, we opt to reduce the weight of the KL divergence term. Even so, KL divergence can still effectively align the distribution of latent variables with the distribution of prior latent variables (see *Figure 1—figure supplement 2*).

In the training stage, we feed raw neural signals into the inference network $f$ to get latent representation $\boldsymbol{z}$, which is regularized by $z_{prior}$ coming from kinematics $\boldsymbol{y}$ and network $m$. Then, we use the mean of $\boldsymbol{z}$, i.e., μ, to decode kinematics $\boldsymbol{y}$ by affine layer $h$ and send $\boldsymbol{z}$ to the generative networks $g$ to generate neural signals $\boldsymbol{x}_r$. To ensure that $\boldsymbol{x}_r$ preserves decoding ability, we send $\boldsymbol{x}_r$ to $f$ and $h$ to decode $\boldsymbol{y}$. The whole model is trained in an end-to-end manner under the guidance of total loss. Once the model has been trained, we can feed raw neural signals to it to obtain behaviorally relevant neural signals $\boldsymbol{x}_r$, and we can also generate behaviorally relevant neural signals using the prior distribution of $z_{prior}$.

## Differences from pi-VAE

pi-VAE lacks the decoding constraint on latent variables (*Equation 4*) and the decoding constraint on generated signals (*Equation 6*).

## Cross-validation evaluation of models

For each model, we use the fivefold cross-validation manner to assess performance. Specifically, we divide the data into five equal folds. In each experiment, we take one fold for testing and use the rest for training (taking three folds as the training set and one as the validation set). The reported performance is averaged over test sets of five experiments. The validation sets are used to choose hyperparameters based on the averaged performance of fivefold validation data. To avoid overfitting, we apply the early stopping strategy. Specifically, we assess the criteria (loss for training distillation methods, $R^2$ for training ANN and KF) on the validation set every epoch in the training process. The model is saved when the model achieves better validation performance than the earlier epochs. If the model cannot increase by 1% of the best performance previously obtained within 10 epochs, we stop the training process.

## The strategy for selecting effective behaviorally relevant signals

As previously mentioned, the hyperparameter $\alpha$ of d-VAE plays a crucial role in balancing the trade-off between reconstruction and decoding loss. Once the appropriate value of $\alpha$ is determined, we can use this value to obtain accurate behaviorally relevant signals for subsequent analysis. To determine the optimal value of $\alpha$, we first enumerated different values of $\alpha$ to guide d-VAE in distilling the behaviorally relevant signals. Next, we used ANN to evaluate the decoding $R^2$ of behaviorally relevant ($D_{re}$) and irrelevant ($D_{ir}$) signals generated by each $\alpha$ value. Finally, we selected the $\alpha$ value with the criteria formula $0.75 \times D_{re} + 0.25 \times (1 - D_{ir})$. The $\alpha$ value that obtained the highest criteria score on the validation set was selected as the optimal value.

Note that we did not use neural similarity between behaviorally relevant and raw signals as a criterion for selecting behaviorally relevant signals. This is because determining the threshold for neural similarity is challenging. However, not using similarity as a criterion does not affect the selection of suitable signals because the decoding performance of behaviorally irrelevant signals can indirectly reflect the degree of similarity between the generated behaviorally relevant signals and the raw signals. Specifically, if the generated behaviorally relevant signals are dissimilar to the raw signals, the behaviorally irrelevant signals will contain many useful signals. In other words, when the neural similarity between behaviorally relevant and raw signals is low, the decoding performance of behaviorally irrelevant signals is high. Therefore, the decoding performance of irrelevant signals is a reasonable alternative to the neural similarity.

Regarding the ratio between $D_{re}$ and $D_{ir}$, any ratio greater than or equal to 3:1 is suitable, and we recommend opting for a higher ratio. This recommendation is based on the observation that when the model is biased toward reconstruction (associated with lower $\alpha$ values), the decoding performance of relevant signals improves as $\alpha$ increases, yet it has not reached saturation. At the same time, the decoding performance of irrelevant signals remains low, but their fluctuations are larger than the improvements in the decoding performance of relevant signals. Consequently, setting the ratio too low poses a risk of selecting an $\alpha$ value where both irrelevant and relevant signals demonstrate low decoding performance. This situation fails to satisfy our requirement that relevant signals should exhibit high decoding performance.

For other generative models, we iterate through a range of hyperparameters, generating the corresponding behaviorally relevant neural signals, and subsequently evaluate these signals using ANN. The hyperparameter associated with the signals that exhibit the highest ANN decoding performance is then selected. In other words, the signals corresponding to this particular hyperparameter are chosen as the selected behaviorally relevant neural signals.

## Implementation details for methods

All the VAE-based models use the Poisson observation function. The details of different methods are demonstrated as follows:

- d-VAE. The encoder $f$ of d-VAE uses two hidden layer MLP with 300 and 100 hidden units. The activation function of the hidden layers is ReLU. The dimensionality of the latent variable is set

to 50. The decoder $g$ of d-VAE is symmetric with the encoder. The last layer of the decoder is followed by a SoftPlus activation function. The prior networks $m$ use one hidden layer MLP with 300 units. The $\beta$ is set to 0.001. The $\alpha$ is set to 0.3, 0.4, 0.7, and 0.9 for datasets A, B, and C. We perform a grid search for $\alpha$ in $\{0.1, 0.2, 0.3, 0.4, 0.5, 0.6, 0.7, 0.8, 0.9\}$, and $\beta$ in $\{0.001, 0.01, 0.1, 1\}$, and latent variable dimension in $\{10, 20, 50\}$. For the synthetic dataset A and B experiments, the $\beta$ and the latent variable dimension are directly set to 0.001 and 50. For synthetic dataset the $\alpha$ is set to 0.9, which is grid searched in $\{0.01\text{–}0.09, 0.1\text{–}0.9, 1\text{–}9\}$, with intervals of 0.01, 0.1, and 1, respectively.

- pi-VAE. The original paper utilizes label information (as shown in *Equation 6*) to approximate the posterior of the latent variable and performs Monte Carlo sampling for decoding during the test stage. However, in our signal generation setting, it is inappropriate to use label information (kinematics) for extracting behaviorally relevant signals during the test stage. As a result, we modified the model to exclude the use of label information in approximating the posterior. The encoder, decoder, and prior networks of our pi-VAE are kept the same as those in d-VAE.

- LFADS. The hidden units of the encoder for the generator's initial conditions, the controller, the generator are set to 200, 200, 200, and 100 for datasets A, B, and C and the synthetic dataset. The dimensionality of latent factor is set to 50 for all datasets. We perform a grid search for hidden units in $\{100, 200\}$, and latent factor dimensions in $\{10, 20, 50\}$. The dimensionality of inferred inputs is set to 1. The Poisson likelihood function is used. The training strategy follows the practice of the original paper. For datasets A and B, a trial length of 18 is set. Trials with lengths below the threshold are zero-padded, while trials exceeding the threshold are truncated to the threshold length from their starting point. In dataset A, there are several trials with lengths considerably longer than that of most trials. We found that padding all trials with zeros to reach the maximum length (32) led to poor performance. Consequently, we chose a trial length of 18, effectively encompassing the durations of most trials and leading to the removal of approximately 9% of samples. For dataset B (center-out), the trial lengths are relatively consistent with small variation, and the maximum length across all trials is 18. For dataset C, we set the trial length as 10 because we observed the video of this paradigm and found that the time for completing a single trial was approximately 1 s. The segments are not overlapped.

- TNDM. The hidden units of the encoder for the generator's initial conditions, the controller, and the generator are set to 64, 64, 100, and 64 for datasets A, B, and C and the synthetic dataset. The dimensionality of relevant latent factors is set to 50, 25, 50, and 50 for datasets A, B, and C and the synthetic dataset. We set the dimensionality of irrelevant latent factors as the same as that of relevant latent factors. The behavior weight is set to 5, 5, 0.2, 0.5 for datasets A, B, and C and the synthetic dataset. We perform a grid search for relevant latent factor dimensionality in $\{10, 25, 50\}$, the hidden units in $\{64, 100, 200\}$, and the behavior weight in $\{0.2, 0.5, 1, 5, 10\}$. The Poisson likelihood function is used. The other hyperparameter setting and the training strategy follow the practice of the original paper. TNDM uses the same trial length and data as LFADS.

- PSID. PSID uses several (horizon size) past neural data to predict behavior at the current time without using neural observations at the current time. For a fair comparison, we let PSID see current neural observations by shifting the neural data one sample into the past relative to the behavior data. We perform a grid search for the horizon hyperparameter in $\{2, 3, 4, 5, 6, 7\}$. Due to the relevant latent dimension should be lower than the horizon times the dimensionality of behavior variables (two-dimensional velocity in this paper), we just set the relevant latent dimension as the maximum. The horizon number of datasets A, B, C, and synthetic datasets is 7, 6, 6, and 5, respectively. And thus the latent variable dimension of datasets A, B, and C and the synthetic dataset is 14, 12, 12, and 10, respectively.

- ANN. ANN has two hidden layers with 300 and 100 hidden units. The activation function of the hidden layers is ReLU.

- KF. The matrix parameters of observation and state transition process are optimized with the least square algorithm. KF is a linear-Gaussian state-space model designed to provide an optimal estimate of the current state (kinematics in this paper). It does so by considering both the current measurement observations (neural signals in this paper) and the previous state estimate. The KF operates in a recursive and iterative manner, continually updating its state estimate as new observations become available.

## Percentage of explained variance captured in a subspace

We applied PCA to behaviorally relevant and irrelevant signals to get relevant PCs and irrelevant PCs. Then, we used the percentage variance captured (also called alignment index) to quantify how many

variances of irrelevant signals can be captured by relevant PCs by projecting irrelevant signals onto the subspace composed of some relevant PCs and vice versa. The percentage of variance captured is

$$A = \frac{\text{Tr}\left(D^{\text{T}} C D\right)}{\text{Tr}\left(C\right)} \times 100\%, \tag{11}$$

where $D \in \mathbb{R}^{N \times d}$ is the top $d$ PCs of relevant signals (irrelevant signals). $C \in \mathbb{R}^{N \times N}$ is the covariance matrix of irrelevant signals (relevant signals), and $\text{Tr}\left(C\right) = \sum_{i=1}^{N} \lambda_i$, where $\lambda_i$ is the $i$th largest eigenvalue for $C$. The percentage variance is a quantity between 0% and 100%.

## The composition of raw signals' variance

Suppose $x \in \mathbb{R}^{1 \times T}$, $y \in \mathbb{R}^{1 \times T}$, and $z \in \mathbb{R}^{1 \times T}$ are the random variables for a single neuron of behaviorally relevant signals, behaviorally irrelevant, and raw signals, where $z = x + y$, $T$ denotes the number of samples. The composition of raw signals' variance is as follows:

$$
\begin{aligned}
\text{Var}(z) &= \text{Var}(x + y) = \mathbb{E}\left\{[x - \mathbb{E}(x)] + [y - \mathbb{E}(y)]\right\}^2 \\
&= \mathbb{E}[x - \mathbb{E}(x)]^2 + \mathbb{E}[y - \mathbb{E}(y)]^2 + 2\mathbb{E}[x - \mathbb{E}(x)][y - \mathbb{E}(y)] \\
&= \text{Var}(x) + \text{Var}(y) + 2\text{Cov}(x, y),
\end{aligned}
\tag{12}
$$

where $\mathbb{E}$, Var, and Cov denote expectation, variance, and covariance, respectively. Thus, the variance of raw signals is composed of the variance of relevant signals $\text{Var}(x)$, the variance of irrelevant signals $\text{Var}(y)$, and the correlation between relevant and irrelevant signals $2\text{Cov}(x, y)$. If there are $N$ neurons, calculate the composition of each neuron separately and then add it up to get the total composition of raw signals.

## Reordered correlation matrix of neurons

The correlation matrix is reordered with a simple group method. The order of reordered neurons is determined from raw neural signals, which is then used for behaviorally relevant signals.

The steps of the group method are as follows:

Step 1: We get the correlation matrix in original neuron order and set a list A that contains all neuron numbers and an empty list B.

Step 2: We select the neuron with the row number of the largest value of the correlation matrix except for the diagonal line and choose nine neurons in list A that are most similar to the selected neuron. We selected the value nine because it offers a good visualization of neuron clusters.

Step 3: Remove these selected neurons from list A, and add these selected neurons in descending order of correlation value in list B.

Step 4: Repeat steps 2 and 3 until list A is empty.

## The improvement ratio of lower- and higher-speed regions
### Split lower- and higher-speed regions

Since the speed ranges of different datasets are different, it is hard to determine a common speed threshold to split lower- and higher-speed regions. Here, we used the squared error as a criterion to split the two speed regions. And for the convenience of calculating the absolute improvement ratio, we need a unified benchmark for comparison. Therefore, we use half of the total squared error as the threshold. Specifically, first, we calculated all samples' total squared error ($E_p$) between actual velocity and predicted velocity obtained by primary signals only. Then, we enumerated the speed from 1 to 50 with a step of 0.1 and calculated the total squared error of selected samples whose speed is less than the enumerated speed. Once the total squared error of selected samples is greater than or equal to the half total squared error of all samples ($0.5E_p$), the enumerated speed is set as the speed threshold. The samples whose speed is less than or equal to the speed threshold belong to lower-speed regions, and those whose speed is greater than the speed threshold belong to higher-speed regions. The squared error of the lower-speed part ($E_p^{low}$) is approximately equal to that of the higher one $E_p^{high}$, i.e., $E_p^{low} \approx E_p^{high} \approx 0.5E_p$ (the difference is negligible).

## The absolute improvement ratio

After splitting the speed regions, we calculated the improvement of the two regions by superimposing secondary signals to primary signals, and got the squared error of lower $E_{p+s}^{low}$ and higher $E_{p+s}^{high}$ regions. Then, we calculated the absolute improvement ratio (AIR):

$$AIR_{low} = -\frac{E_{p+s}^{low} - E_p^{low}}{E_p^{low}} \approx -\frac{E_{p+s}^{low} - 0.5E_p}{0.5E_p} \tag{13}$$

$$AIR_{high} = -\frac{E_{p+s}^{high} - E_p^{high}}{E_p^{high}} \approx -\frac{E_{p+s}^{high} - 0.5E_p}{0.5E_p} \tag{14}$$

Since the two regions refer to a common standard ($0.5E_p$), the improvement ratio of the two regions can be directly compared, and that's why we call it the absolute improvement ratio.

## The relative improvement ratio

The relative improvement ratio (RIR) measures the improvement ratio of each sample relative to itself before and after superimposing secondary signals. The relative improvement ratio is computed as follows:

$$RIR^i = -\frac{E_{p+s}^i - E_p^i}{E_p^i}, \tag{15}$$

where $i$ denotes the $i$th sample of test data.

## Code availability

The code is available at https://github.com/eric0li/d-VAE, copy archived at *eric0li, 2024*.

## Acknowledgements

We thank Yiwen Wang for sharing the monkey obstacle avoidance reaching data. We thank Yuxiao Yang and Huaqin Sun for valuable discussions. This work was supported in part by the National Natural Science Foundation of China under Grants 62336007, in part by the Key R&D Program of Zhejiang under Grant 2022C03011, in part by the Starry Night Science Fund of Zhejiang University Shanghai Institute for Advanced Study under Grant SN-ZJU-SIAS-002, and in part by the Fundamental Research Funds for the Central Universities.

## Additional information

### Competing interests

Yueming Wang: The other authors declare that no competing interests exist.

### Funding

| Funder | Grant reference number | Author |
| --- | --- | --- |
| National Natural Science Foundation of China | no. 62336007 | Yueming Wang |
| Key Research and Development Program of Zhejiang | no. 2022C03011 | Yueming Wang |
| Starry Night Science Fund of Zhejiang University Shanghai Institute for Advanced Study | SN-ZJU-SIAS-002 | Yueming Wang |

| Funder | Grant reference number | Author |
| --- | --- | --- |
| Fundamental Research Funds for the Central Universities | | Yueming Wang |

The funders had no role in study design, data collection and interpretation, or the decision to submit the work for publication.

## Author contributions

Yangang Li, Conceptualization, Software, Formal analysis, Visualization, Methodology, Writing – original draft, Writing – review and editing; Xinyun Zhu, Visualization, Writing – original draft, Writing – review and editing; Yu Qi, Supervision, Writing – original draft; Yueming Wang, Supervision, Funding acquisition, Writing – review and editing

## Author ORCIDs

Yangang Li ⬤ https://orcid.org/0000-0002-7271-2993
Xinyun Zhu ⬤ https://orcid.org/0009-0007-3820-4761
Yueming Wang ⬤ https://orcid.org/0000-0001-7742-0722

## Ethics

All animal handling procedures were authorized by the Animal Care Committee at Zhejiang University, China (No. ZJU20220142), and conducted following the Guide for Care and Use of Laboratory Animals (China Ministry of Health).

Reviewer #1 (Public Review): https://doi.org/10.7554/eLife.87881.4.sa1
Reviewer #2 (Public Review): https://doi.org/10.7554/eLife.87881.4.sa2
Reviewer #4 (Public Review): https://doi.org/10.7554/eLife.87881.4.sa3
Author response https://doi.org/10.7554/eLife.87881.4.sa4

# Additional files

## Supplementary files

• MDAR checklist

## Data availability

All the datasets are publicly accessible.

The following previously published dataset was used:

| Author(s) | Year | Dataset title | Dataset URL | Database and Identifier |
| --- | --- | --- | --- | --- |
| O'Doherty JE, Cardoso MMB, Makin JG, Sabes PN | 2017 | Nonhuman Primate Reaching with Multichannel Sensorimotor Cortex Electrophysiology: broadband for indy_20170124_01 | https://doi.org/10.5281/zenodo.1163026 | Zenodo, 10.5281/zenodo.1163026 |

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
