## [Editor Report · eLife assessment]

This study presents a **useful** method for the extraction of behaviour-related activity from neural population recordings based on a specific deep learning architecture, a variational autoencoder. Although the authors performed thorough benchmarking of their method in the context of decoding behavioural variables, the evidence supporting claims about encoding is **incomplete** as the results may stem, in part, from the properties of the method itself.

---

## [Referee Report · Reviewer #1 (Public Review)]

This work seeks to understand how behaviour-related information is represented in the neural activity of the primate motor cortex. To this end, a statistical model of neural activity is presented that enables a non-linear separation of behaviour-related from unrelated activity. As a generative model, it enables the separate analysis of these two activity modes, here primarily done by assessing the decoding performance of hand movements the monkeys perform in the experiments. Several lines of analysis are presented to show that while the neurons with significant tuning to movements strongly contribute to the behaviourally-relevant activity subspace, less or un-tuned neurons also carry decodable information. It is further shown that the discovered subspaces enable linear decoding, leading the authors to conclude that motor cortex read-out can be linear.

Strengths:

In my opinion, using an expressive generative model to analyse neural state spaces is an interesting approach to understand neural population coding. While potentially sacrificing interpretability, this approach allows capturing both redundancies and synergies in the code as done in this paper. The model presented here is a natural non-linear extension of a previous linear model (PSID) and uses weak supervision in a manner similar to a previous non-linear model (TNDM).

Weaknesses:

This revised version provides additional evidence to support the author's claims regarding model performance and interpretation of the structure of the resulting latent spaces, in particular the distributed neural code over the whole recorded population, not just the well-tuned neurons. The improved ability to linearly decode behaviour from the relevant subspace and the analysis of the linear subspace projections in my opinion convincingly demonstrates that the model picks up behaviour-relevant dynamics, and that these are distributed widely across the population. As reviewer 3 also points out, I would, however, caution to interpret this as evidence for linear read-out of the motor system - your model performs a non-linear transformation, and while this is indeed linearly decodable, the motor system would need to do something similar first to achieve the same. In fact to me it seems to show the opposite, that behaviour-related information may not be generally accessible to linear decoders (including to down-stream brain areas).

As in my initial review, I would also caution against making strong claims about identifiability although this work and TNDM seem to show that in practise such methods work quite well. CEBRA, in contrast, offers some theoretical guarantees, but it is not a generative model, so would not allow the type of analysis done in this paper. In your model there is a para,eter \alpha to balance between neural and behaviour reconstruction. This seems very similar to TNDM and has to be optimised - if this is correct, then there is manual intervention required to identify a good model.

Somewhat related, I also found that the now comprehensive comparison with related models shows that the using decoding performance (R2) as a metric for model comparison may be problematic: the R2 values reported in Figure 2 (e.g. the MC_RTT dataset) should be compared to the values reported in the neural latent benchmark, which represent well-tuned models (e.g. AutoLFADS). The numbers (difficult to see, a table with numbers in the appendix would be useful, see: https://eval.ai/web/challenges/challenge-page/1256/leaderboard) seem lower than what can be obtained with models without latent space disentanglement. While this does not necessarily invalidate the conclusions drawn here, it shows that decoding performance can depend on a variety of model choices, and may not be ideal to discriminate between models. I'm also surprised by the low neural R2 for LFADS I assume this is condition-averaged - LFADS tends to perform very well on this metric.

One statement I still cannot follow is how the prior of the variational distribution is modelled. You say you depart from the usual Gaussian prior, but equation 7 seems to suggest there is a normal prior. Are the parameters of this distribution learned? As I pointed out earlier, I however suspect this may not matter much as you give the prior a very low weight. I also still am not sure how you generate a sample from the variational distribution, do you just draw one for each pass?

Summary:

This paper presents a very interesting analysis, but some concerns remain that mainly stem from the complexity of deep learning models. It would be good to acknowledge these as readers without relevant background need to understand where the possible caveats are.

---

## [Referee Report · Reviewer #2 (Public Review)]

Li et al present a method to extract "behaviorally relevant" signals from neural activity. The method is meant to solve a problem which likely has high utility for neuroscience researchers. There are numerous existing methods to achieve this goal some of which the authors compare their method to-thankfully, the revised version includes one of the major previous omissions (TNDM). However, I still believe that d-VAE is a promising approach that has its own advantages. Still, I have issues with the paper as-is. The authors have made relatively few modifications to the text based on my previous comments, and the responses have largely just dismissed my feedback and restated claims from the paper. Nearly all of my previous comments remain relevant for this revised manuscript. As such, they have done little to assuage my concerns, the most important of which I will restate here using the labels/notation (Q1, Q2, etc) from the reviewer response.

(Q1) I still remain unconvinced that the core findings of the paper are "unexpected". In the response to my previous Specific Comment #1, they say "We use the term 'unexpected' due to the disparity between our findings and the prior understanding concerning neural encoding and decoding." However, they provide no citations or grounding for why they make those claims. What prior understanding makes it unexpected that encoding is more complex than decoding given the entropy, sparseness, and high dimensionality of neural signals (the "encoding") compared to the smoothness and low dimensionality of typical behavioural signals (the "decoding")?

(Q2) I still take issue with the premise that signals in the brain are "irrelevant" simply because they do not correlate with a fixed temporal lag with a particular behavioural feature hand-chosen by the experimenter. In the response to my previous review, the authors say "we employ terms like 'behaviorally-relevant' and 'behaviorally-irrelevant' only regarding behavioral variables of interest measured within a given task, such as arm kinematics during a motor control task.". This is just a restatement of their definition, not a response to my concern, and does not address my concern that the method requires a fixed temporal lag and continual decoding/encoding. My example of reward signals remains. There is a huge body of literature dating back to the 70s on the linear relationships between neural and activity and arm kinematics; in a sense, the authors have chosen the "variable of interest" that proves their point. This all ties back to the previous comment: this is mostly expected, not unexpected, when relating apparently-stochastic, discrete action potential events to smoothly varying limb kinematics.

(Q5) The authors seem to have missed the spirit of my critique: to say "linear readout is performed in motor cortex" is an over-interpretation of what their model can show.

(Q7) Agreeing with my critique is not sufficient; please provide the data or simulations that provides the context for the reference in the fano factor. I believe my critique is still valid.

(Q8) Thank you for comparing to TNDM, it's a useful benchmark.

---

## [Referee Report · Reviewer #4 (Public Review)]

I am a new reviewer for this manuscript, which has been reviewed before. The authors provide a variational autoencoder that has three objectives in the loss: linear reconstruction of behavior from embeddings, reconstruction of neural data, and KL divergence term related to the variational model elements. They take the output of the VAE as the "behaviorally relevant" part of neural data and call the residual "behaviorally irrelevant". Results aim to inspect the linear versus nonlinear behavior decoding using the original raw neural data versus the inferred behaviorally relevant and irrelevant parts of the signal.

Overall, studying neural computations that are behaviorally relevant or not is an important problem, which several previous studies have explored (for example PSID in (Sani et al. 2021), TNDM in (Hurwitz et al. 2021), TAME-GP in (Balzani et al. 2023), pi-VAE in (Zhou and Wei 2020), and dPCA in (Kobak et al. 2016), etc). However, this manuscript does not properly put their work in the context of such prior works. For example, the abstract states "One solution is to accurately separate behaviorally-relevant and irrelevant signals, but this approach remains elusive", which is not the case given that these prior works have done that. The same is true for various claims in the main text, for example "Furthermore, we found that the dimensionality of primary subspace of raw signals (26, 64, and 45 for datasets A, B, and C) is significantly higher than that of behaviorally-relevant signals (7, 13, and 9), indicating that using raw signals to estimate the neural dimensionality of behaviors leads to an overestimation" (line 321). This finding was presented in (Sani et al. 2021) and (Hurwitz et al. 2021), which is not clarified here. This issue of putting the work in context has been brought up by other reviewers previously but seems to remain largely unaddressed. The introduction is inaccurate also in that it mixes up methods that were designed for separation of behaviorally relevant information with those that are unsupervised and do not aim to do so (e.g., LFADS). The introduction should be significantly revised to explicitly discuss prior models/works that specifically formulated this behavior separation and what these prior studies found, and how this study differs.

Beyond the above, some of the main claims/conclusions made by the manuscript are not properly supported by the analyses and results, which has also been brought up by other reviewers but not fully addressed. First, the analyses here do not support the linear readout from the motor cortex because (i) by construction, the VAE here is trained to have a linear readout from its embedding in its loss, which can bias its outputs toward doing well with a linear decoder/readout, and (ii) the overall mapping from neural data to behavior includes both the VAE and the linear readout and thus is always nonlinear (even when a linear Kalman filter is used for decoding). This claim is also vague as there is no definition of readout from "motor cortex" or what it means. Why is the readout from the bottleneck of this particular VAE the readout of motor cortex? Second, other claims about properties of individual neurons are also confounded because the VAE is a population-level model that extracts the bottleneck from all neurons. Thus, information can leak from any set of neurons to other sets of neurons during the inference of behaviorally relevant parts of signals. Overall, the results do not convincingly support the claims, and thus the claims should be carefully revised and significantly tempered to avoid misinterpretation by readers.

Below I briefly expand on these as well as other issues, and provide suggestions:

(1) Claims about linearity of "motor cortex" readout are not supported by results yet stated even in the abstract. Instead, what the results support is that for decoding behavior from the output of the dVAE model -- that is trained specifically to have a linear behavior readout from its embedding -- a nonlinear readout does not help. This result can be biased by the very construction of the dVAE's loss that encourages a linear readout/decoding from embeddings and thus does not imply a finding about motor cortex.

(2) Related to the above, it is unclear what the manuscript means by readout from motor cortex. A clearer definition of "readout" (a mapping from what to what?) in general is needed. The mapping that the linearity/nonlinearity claims refer to is from the *inferred* behaviorally relevant neural signals, which themselves are inferred nonlinearly using the VAE. This should be explicitly clarified in all claims, i.e., that only the mapping from distilled signals to behavior is linear, not the whole mapping from neural data to behavior. Again, to say the readout from motor cortex is linear is not supported, including in the abstract.

(3) Claims about individual neurons are also confounded. The d-VAE distilling processing is a population level embedding so the individual distilled neurons are not obtainable on their own without using the population data. This population level approach also raises the possibility that information can leak from one neuron to another during distillation, which is indeed what the authors hope would recover true information about individual neurons that wasn't there in the recording (the pixel denoising example). The authors acknowledge the possibility that information could leak to a neuron that didn't truly have that information and try to rule it out to some extent with some simulations and by comparing the distilled behaviorally relevant signals to the original neural signals. But ultimately, the distilled signals are different enough from the original signals to substantially improve decoding of low information neurons, and one cannot be sure if all of the information in distilled signals from any individual neuron truly belongs to that neuron. It is still quite likely that some of the improved behavior prediction of the distilled version of low-information neurons is due to leakage of behaviorally relevant information from other neurons, not the former's inherent behavioral information. This should be explicitly acknowledged in the manuscript.

(4) Given the nuances involved in appropriate comparisons across methods and since two of the datasets are public, the authors should provide their complete code (not just the dVAE method code), including the code for data loading, data preprocessing, model fitting and model evaluation for all methods and public datasets. This will alleviate concerns and allow readers to confirm conclusions (e.g., figure 2) for themselves down the line.

(5) Related to (1) above, the authors should explore the results if the affine network h(.) (from embedding to behavior) was replaced with a nonlinear ANN. Perhaps linear decoders would no longer be as close to nonlinear decoders. Regardless, the claim of linearity should be revised as described in (1) and (2) above, and all caveats should be discussed.

(6) The beginning of the section on the "smaller R2 neurons" should clearly define what R2 is being discussed. Based on the response to previous reviewers, this R2 "signifies the proportion of neuronal activity variance explained by the linear encoding model, calculated using raw signals". This should be mentioned and made clear in the main text whenever this R2 is referred to.

(7) Various terms require clear definitions. The authors sometimes use vague terminology (e.g., "useless") without a clear definition. Similarly, discussions regarding dimensionality could benefit from more precise definitions. How is neural dimensionality defined? For example, how is "neural dimensionality of specific behaviors" (line 590) defined? Related to this, I agree with Reviewer 2 that a clear definition of irrelevant should be mentioned that clarifies that relevance is roughly taken as "correlated or predictive with a fixed time lag". The analyses do not explore relevance with arbitrary time lags between neural and behavior data.

(8) CEBRA itself doesn't provide a neural reconstruction from its embeddings, but one could obtain one via a regression from extracted CEBRA embeddings to neural data. In addition to decoding results of CEBRA (figure S3), the neural reconstruction of CEBRA should be computed and CEBRA should be added to Figure 2 to see how the behaviorally relevant and irrelevant signals from CEBRA compare to other methods.

References:

Kobak, Dmitry, Wieland Brendel, Christos Constantinidis, Claudia E Feierstein, Adam Kepecs, Zachary F Mainen, Xue-Lian Qi, Ranulfo Romo, Naoshige Uchida, and Christian K Machens. 2016. "Demixed Principal Component Analysis of Neural Population Data." Edited by Mark CW van Rossum. eLife 5 (April): e10989. https://doi.org/10.7554/eLife.10989.

Sani, Omid G., Hamidreza Abbaspourazad, Yan T. Wong, Bijan Pesaran, and Maryam M. Shanechi. 2021. "Modeling Behaviorally Relevant Neural Dynamics Enabled by Preferential Subspace Identification." Nature Neuroscience 24 (1): 140-49. https://doi.org/10.1038/s41593-020-00733-0.

Zhou, Ding, and Xue-Xin Wei. 2020. "Learning Identifiable and Interpretable Latent Models of High-Dimensional Neural Activity Using Pi-VAE." In Advances in Neural Information Processing Systems, 33:7234-47. Curran Associates, Inc https://proceedings.neurips.cc/paper/2020/hash/510f2318f324cf07fce24c3a4b89c771-Abstract.html.

Hurwitz, Cole, Akash Srivastava, Kai Xu, Justin Jude, Matthew Perich, Lee Miller, and Matthias Hennig. 2021. "Targeted Neural Dynamical Modeling." In Advances in Neural Information Processing Systems. Vol. 34. https://proceedings.neurips.cc/paper/2021/hash/f5cfbc876972bd0d031c8abc37344c28-Abstract.html.

Balzani, Edoardo, Jean-Paul G. Noel, Pedro Herrero-Vidal, Dora E. Angelaki, and Cristina Savin. 2023. "A Probabilistic Framework for Task-Aligned Intra- and Inter-Area Neural Manifold Estimation." In . https://openreview.net/forum?id=kt-dcBQcSA.

---

## [Author Response]

The following is the authors’ response to the previous reviews.

To the Senior Editor and the Reviewing Editor:

We sincerely appreciate the valuable comments provided by the reviewers, the reviewing editor, and the senior editor. Based on our last response and revision, we are confused by the two limitations noted in the eLife assessment.

(1) benchmarking against comparable methods is limited.

In our last revision, we added the comparison experiments with TNDM, as the reviewers requested. Additionally, it is crucial to emphasize that our evaluation of decoding capabilities of behaviorally relevant signals has been benchmarked against the performance of the ANN on raw signals, which, as Reviewer #1 previously noted, nearly represents the upper limit of performance. Consequently, we believe that our benchmarking methods are sufficiently strong.

(2) some observations may be a byproduct of their method, and may not constitute new scientific observations.

We believe that our experimental results are sufficient to demonstrate that our conclusions are not byproducts of d-VAE based on three reasons:

(1) The d-VAE, as a latent variable model, adheres to the population doctrine, which posits that latent variables are responsible for generating the activities of individual neurons. The goal of such models is to maximize the explanation of the raw signals. At the signal level, the only criterion we can rely on is neural reconstruction performance, in which we have achieved unparalleled results. Thus, it is inappropriate to focus on the mixing process during the model's inference stage while overlooking the crucial de-mixing process during the generation stage and dismissing the significance of our neural reconstruction results. For more details, please refer to the first point in our response to Q4 from Reviewer #4.

(2) The criterion that irrelevant signals should contain minimal information can effectively demonstrate that our conclusions are not by-products of d-VAE. Unfortunately, the reviewers seem to have overlooked this criterion. For more details, please refer to the third point in our response to Q4 from Reviewer #4

(3) Our synthetic experimental results also substantiate that our conclusions are not byproducts of d-VAE. However, it appears the reviewers did not give these results adequate consideration. For more details, please refer to the fourth point in our response to Q4 from Reviewer #4.

Furthermore, our work presents not just "a useful method" but a comprehensive framework. Our study proposes, for the first time, a framework for defining, extracting, and validating behaviorally relevant signals. In our current revision, to clearly distinguish between d-VAE and other methods, we have formalized the extraction of behaviorally relevant signals into a mathematical optimization problem. To our knowledge, current methods have not explicitly proposed extracting behaviorally relevant signals, nor have they identified and addressed the key challenges of extracting relevant signals. Similarly, existing research has not yet defined and validated behaviorally relevant signals. For more details, please refer to our response to Q1 from Reviewer #4.

Based on these considerations, we respectfully request that you reconsider the eLife assessment of our work. We greatly appreciate your time and attention to this matter.

The main revisions made to the manuscript are as follows:

(1) We have formalized the extraction of behaviorally relevant signals into a mathematical optimization problem, enabling a clearer distinction between d-VAE and other models.

(2) We have moderated the assertion about linear readout to highlight its conjectural nature and have broadened the discussion regarding this conclusion.

(3) We have elaborated on the model details of d-VAE and have removed the identifiability claim.

To Reviewer #1

Q1: “As reviewer 3 also points out, I would, however, caution to interpret this as evidence for linear read-out of the motor system - your model performs a non-linear transformation, and while this is indeed linearly decodable, the motor system would need to do something similar first to achieve the same. In fact to me it seems to show the opposite, that behaviour-related information may not be generally accessible to linear decoders (including to down-stream brain areas).”

Thank you for your comments. It's important to note that the conclusions we draw are speculative and not definitive. We use terms like "suggest" to reflect this uncertainty. To further emphasize the conjectural nature of our conclusions, we have deliberately moderated our tone.

The question of whether behaviorally-relevant signals can be accessed by linear decoders or downstream brain regions hinges on the debate over whether the brain employs a strategy of filtering before decoding. If the brain employs such a strategy, the brain can probably access these signals. In our opinion, it is likely that the brain utilizes this strategy.

Given the existence of behaviorally relevant signals, it is reasonable to assume that the brain has intrinsic mechanisms to differentiate between relevant and irrelevant signals. There is growing evidence suggesting that the brain utilizes various mechanisms, such as attention and specialized filtering, to suppress irrelevant signals and enhance relevant signals [1-3]. Therefore, it is plausible that the brain filters before decoding, thereby effectively accessing behaviorally relevant signals.

Thank you for your valuable feedback.

(1) Sreenivasan, Sameet, and Ila Fiete. "Grid cells generate an analog error-correcting code for singularly precise neural computation." Nature neuroscience 14.10 (2011): 1330-1337.

(2) Schneider, David M., Janani Sundararajan, and Richard Mooney. "A cortical filter that learns to suppress the acoustic consequences of movement." Nature 561.7723 (2018): 391-395.

(3) Nakajima, Miho, L. Ian Schmitt, and Michael M. Halassa. "Prefrontal cortex regulates sensory filtering through a basal ganglia-to-thalamus pathway." Neuron 103.3 (2019): 445-458.

Q2: “As in my initial review, I would also caution against making strong claims about identifiability although this work and TNDM seem to show that in practise such methods work quite well. CEBRA, in contrast, offers some theoretical guarantees, but it is not a generative model, so would not allow the type of analysis done in this paper. In your model there is a para,eter \alpha to balance between neural and behaviour reconstruction. This seems very similar to TNDM and has to be optimised - if this is correct, then there is manual intervention required to identify a good model.”

Thank you for your comments.

Considering your concerns about our identifiability claims and the fact that identifiability is not directly relevant to the core of our paper, we have removed content related to identifiability.

Firstly, our model is based on the pi-VAE, which also has theoretical guarantees. However, it is important to note that all such theoretical guarantees (including pi-VAE and CEBRA) are based on certain assumptions that cannot be validated as the true distribution of latent variables remains unknown.

Secondly, it is important to clarify that the identifiability of latent variables does not impact the conclusions of this paper, nor does this paper make specific conclusions about the model's latent variables. Identifiability means that distinct latent variables correspond to distinct observations. If multiple latent variables can generate the same observation, it becomes impossible to determine which one is correct given the observation, which leads to the issue of nonidentifiability. Notably, our analysis focuses on the generated signals, not the latent variables themselves, and thus the identifiability of these variables does not affect our findings.

Our approach, dedicated to extracting these signals, distinctly differs from methods such as TNDM, which focuses on extracting behaviorally relevant latent dynamics. To clearly set apart d-VAE from other models, we have framed the extraction of behaviorally relevant signals as the following mathematical optimization problem:minxrE(xr,x)+R(xr)

where 𝑥# denotes generated behaviorally-relevant signals, 𝑥 denotes raw noisy signals, 𝐸(⋅,⋅) demotes reconstruction loss, and 𝑅(⋅) denotes regularization loss. It is important to note that while both d-VAE and TNDM employ reconstruction loss, relying solely on this term is insufficient for determining the optimal degree of similarity between the generated and raw noisy signals. The key to accurately extracting behaviorally relevant signals lies in leveraging prior knowledge about these signals to determine the optimal similarity degree, encapsulated by 𝑅(𝒙𝒓). Other studies have not explicitly proposed extracting behaviorally-relevant signals, nor have they identified and addressed the key challenges involved in extracting relevant signals. Consequently, our approach is distinct from other methods.

Thank you for your valuable feedback.

Q3: “Somewhat related, I also found that the now comprehensive comparison with related models shows that the using decoding performance (R2) as a metric for model comparison may be problematic: the R2 values reported in Figure 2 (e.g. the MC_RTT dataset) should be compared to the values reported in the neural latent benchmark, which represent well-tuned models (e.g. AutoLFADS). The numbers (difficult to see, a table with numbers in the appendix would be useful, see: https://eval.ai/web/challenges/challenge-page/1256/leaderboard) seem lower than what can be obtained with models without latent space disentanglement. While this does not necessarily invalidate the conclusions drawn here, it shows that decoding performance can depend on a variety of model choices, and may not be ideal to discriminate between models. I'm also surprised by the low neural R2 for LFADS I assume this is condition-averaged - LFADS tends to perform very well on this metric.”

Thank you for your comments. The dataset we utilized is not from the same day as the neural latent benchmark dataset. Notably, there is considerable variation in the length of trials within the RTT paradigm, and the dataset lacks explicit trial information, rendering trial-averaging unsuitable. Furthermore, behaviorally relevant signals are not static averages devoid of variability; even behavioral data exhibits variability. We computed the neural R2 using individual trials rather than condition-averaged responses.

Thank you for your valuable feedback.

Q4: “One statement I still cannot follow is how the prior of the variational distribution is modelled. You say you depart from the usual Gaussian prior, but equation 7 seems to suggest there is a normal prior. Are the parameters of this distribution learned? As I pointed out earlier, I however suspect this may not matter much as you give the prior a very low weight. I also still am not sure how you generate a sample from the variational distribution, do you just draw one for each pass?”

Thank you for your questions.

The conditional distribution of prior latent variables 𝑝%(𝒛|𝒚) is a Gaussian distribution, but the distribution of prior latent variables 𝑝(𝒛) is a mixture Gaussian distribution. The distribution of prior latent variables 𝑝(𝒛) is:p(z)=∫p^(y)pm(z∣y)dy=∑j=1N(1N∑i=1Nδ(y(j)−y(i)))pm(z∣y(j))

where p^(y)=1 N∑i=1Nδ(y−y(i)) denotes the empirical distribution of behavioral variables

𝒚, and 𝑁 denotes the number of samples, 𝒚(𝒊) denotes the 𝒊th sample, δ(⋅) denotes the Dirac delta function, and 𝑝%(𝒛|𝒚) denotes the conditional distribution of prior latent variables given the behavioral variables parameterized by network 𝑚. Based on the above equation, we can see that 𝑝(𝒛) is not a Gaussian distribution, it is a Gaussian mixture model with 𝑁 components, which is theoretically a universal approximator of continuous probability densities.

Learning this prior is important, as illustrated by our latent variable visualizations, which are not a Gaussian distribution. Upon conducting hypothesis testing for both latent variables and behavioral variables, neither conforms to Gaussian distribution (Lilliefors test and Kolmogorov-Smirnov test). Consequently, imposing a constraint on the latent variables towards N(0,1) is expected to affect performance adversely.

Regarding sampling, during training process, we draw only one sample from the approximate posterior distribution q(z(i)∣x(i)) . It is worth noting that drawing multiple samples or one sample for each pass does not affect the experimental results. After training, we can generate a sample from the prior by providing input behavioral data 𝒚(𝒊) and then generating corresponding samples via p(z(i)∣y(i)) and p(x(i)∣z(i)) . To extract behaviorally-relevant signals from raw signals, we use q(z(i)∣x(i)) and p(x(i)∣z(i)) .

Thank you for your valuable feedback.

Q5: “(1) I found the figures good and useful, but the text is, in places, not easy to follow. I think the manuscript could be shortened somewhat, and in some places more concise focussed explanations would improve readability.(2) I would not call the encoding "complex non-linear" - non-linear is a clear term, but complex can mean many things (e.g. is a quadratic function complex?) ”

Thank you for your recommendation. We have revised the manuscript for enhanced clarity. We call the encoding “complex nonlinear” because neurons encode information with varying degrees of nonlinearity, as illustrated in Fig. 3b, f, and Fig. S3b.

Thank you for your valuable feedback.

To Reviewer #2

Q1: “I still remain unconvinced that the core findings of the paper are "unexpected". In the response to my previous Specific Comment #1, they say "We use the term 'unexpected' due to the disparity between our findings and the prior understanding concerning neural encoding and decoding." However, they provide no citations or grounding for why they make those claims. What prior understanding makes it unexpected that encoding is more complex than decoding given the entropy, sparseness, and high dimensionality of neural signals (the "encoding") compared to the smoothness and low dimensionality of typical behavioural signals (the "decoding")?”

Thank you for your comments. We believe that both the complexity of neural encoding and the simplicity of neural decoding in motor cortex are unexpected.

The Complexity of Neural Encoding: As noted in the Introduction, neurons with small R2 values were traditionally considered noise and consequently disregarded, as detailed in references [1-3]. However, after filtering out irrelevant signals, we discovered that these neurons actually contain substantial amounts of behavioral information, previously unrecognized. Similarly, in population-level analyses, neural signals composed of small principal components (PCs) are often dismissed as noise, with analyses typically utilizing only between 6 and 18 PCs [4-10]. Yet, the discarded PC signals nonlinearly encode significant amounts of information, with practically useful dimensions found to range between 30 and 40—far exceeding the usual number analyzed. These findings underscore the complexity of neural encoding and are unexpected.

The Simplicity of Neural Decoding: In the motor cortex, nonlinear decoding of raw signals has been shown to significantly outperform linear decoding, as evidenced in references [11,12]. Interestingly, after separating behaviorally relevant and irrelevant signals, we observed that the linear decoding performance of behaviorally relevant signals is nearly equivalent to that of nonlinear decoding—a phenomenon previously undocumented in the motor cortex. This discovery is also unexpected.

Thank you for your valuable feedback.

(1) Georgopoulos, Apostolos P., Andrew B. Schwartz, and Ronald E. Kettner. "Neuronal population coding of movement direction." Science 233.4771 (1986): 1416-1419.

(2) Hochberg, Leigh R., et al. "Reach and grasp by people with tetraplegia using a neurally controlled robotic arm." Nature 485.7398 (2012): 372-375.

(3) Inoue, Yoh, et al. "Decoding arm speed during reaching." Nature communications 9.1 (2018): 5243.

(4) Churchland, Mark M., et al. "Neural population dynamics during reaching." Nature 487.7405 (2012): 51-56.

(5) Kaufman, Matthew T., et al. "Cortical activity in the null space: permitting preparation without movement." Nature neuroscience 17.3 (2014): 440-448.

(6) Elsayed, Gamaleldin F., et al. "Reorganization between preparatory and movement population responses in motor cortex." Nature communications 7.1 (2016): 13239.

(7) Sadtler, Patrick T., et al. "Neural constraints on learning." Nature 512.7515 (2014): 423426.

(8) Golub, Matthew D., et al. "Learning by neural reassociation." Nature neuroscience 21.4 (2018): 607-616.

(9) Gallego, Juan A., et al. "Cortical population activity within a preserved neural manifold underlies multiple motor behaviors." Nature communications 9.1 (2018): 4233.

(10) Gallego, Juan A., et al. "Long-term stability of cortical population dynamics underlying consistent behavior." Nature neuroscience 23.2 (2020): 260-270.

(11) Glaser, Joshua I., et al. "Machine learning for neural decoding." Eneuro 7.4 (2020).

(12) Willsey, Matthew S., et al. "Real-time brain-machine interface in non-human primates achieves high-velocity prosthetic finger movements using a shallow feedforward neural network decoder." Nature Communications 13.1 (2022): 6899.

Q2: “I still take issue with the premise that signals in the brain are "irrelevant" simply because they do not correlate with a fixed temporal lag with a particular behavioural feature handchosen by the experimenter. In the response to my previous review, the authors say "we employ terms like 'behaviorally-relevant' and 'behaviorally-irrelevant' only regarding behavioral variables of interest measured within a given task, such as arm kinematics during a motor control task.". This is just a restatement of their definition, not a response to my concern, and does not address my concern that the method requires a fixed temporal lag and continual decoding/encoding. My example of reward signals remains. There is a huge body of literature dating back to the 70s on the linear relationships between neural and activity and arm kinematics; in a sense, the authors have chosen the "variable of interest" that proves their point. This all ties back to the previous comment: this is mostly expected, not unexpected, when relating apparently-stochastic, discrete action potential events to smoothly varying limb kinematics.”

Thank you for your comments.

Regarding the experimenter's specification of behavioral variables of interest, we followed common practice in existing studies [1, 2]. Regarding the use of fixed temporal lags, we followed the same practice as papers related to the dataset we use, which assume fixed temporal lags [3-5]. Furthermore, many studies in the motor cortex similarly use fixed temporal lags [68].

Concerning the issue of rewards, in the paper you mentioned [9], the impact of rewards occurs after the reaching phase. It's important to note that in our experiments, we analyze only the reaching phase, without any post-movement phase.

If the impact of rewards can be stably reflected in the signals in the reaching phase of the subsequent trial, and if the reward-induced signals do not interfere with decoding—since these signals are harmless for decoding and beneficial for reconstruction—our model is likely to capture these signals. If the signals induced by rewards during the reaching phase are randomly unstable, our model will likely be unable to capture them.

If the goal is to extract post-movement neural activity from both rewarded and unrewarded trials, and if the neural patterns differ between these conditions, one could replace the d-VAE's regression loss, used for continuous kinematics decoding, with a classification loss tailored to distinguish between rewarded and unrewarded conditions.

To clarify the definition, we have revised it in the manuscript. Specifically, before a specific definition, we briefly introduce the relevant signals and irrelevant signals. Behaviorally irrelevant signals refer to those not directly associated with the behavioral variables of interest and may include noise or signals from variables of no interest. In contrast, behaviorally relevant signals refer to those directly related to the behavioral variables of interest. For instance, rewards in the post-movement phase are not directly related to behavioral variables (kinematics) in the reaching movement phase.

It is important to note that our definition of behaviorally relevant signals not only includes decoding capabilities but also specific requirement at the signal level, based on two key requirements:

(1) they should closely resemble raw signals to preserve the underlying neuronal properties without becoming so similar that they include irrelevant signals. (encoding requirement), and (2) they should contain behavioral information as much as possible (decoding requirement). Signals that meet both requirements are considered effective behaviorally relevant signals. In our study, we assume raw signals are additively composed of behaviorally-relevant and irrelevant signals. We define irrelevant signals as those remaining after subtracting relevant signals from raw signals. Therefore, we believe our definition is clearly articulated.

Thank you for your valuable feedback.

(1) Sani, Omid G., et al. "Modeling behaviorally relevant neural dynamics enabled by preferential subspace identification." Nature Neuroscience 24.1 (2021): 140-149.

(2) Buetfering, Christina, et al. "Behaviorally relevant decision coding in primary somatosensory cortex neurons." Nature neuroscience 25.9 (2022): 1225-1236.

(3) Wang, Fang, et al. "Quantized attention-gated kernel reinforcement learning for brain– machine interface decoding." IEEE transactions on neural networks and learning systems 28.4 (2015): 873-886.

(4) Dyer, Eva L., et al. "A cryptography-based approach for movement decoding." Nature biomedical engineering 1.12 (2017): 967-976.

(5) Ahmadi, Nur, Timothy G. Constandinou, and Christos-Savvas Bouganis. "Robust and accurate decoding of hand kinematics from entire spiking activity using deep learning." Journal of Neural Engineering 18.2 (2021): 026011.

(6) Churchland, Mark M., et al. "Neural population dynamics during reaching." Nature 487.7405 (2012): 51-56.

(7) Kaufman, Matthew T., et al. "Cortical activity in the null space: permitting preparation without movement." Nature neuroscience 17.3 (2014): 440-448.

(8) Elsayed, Gamaleldin F., et al. "Reorganization between preparatory and movement population responses in motor cortex." Nature communications 7.1 (2016): 13239.

(9) Ramkumar, Pavan, et al. "Premotor and motor cortices encode reward." PloS one 11.8 (2016): e0160851.

Q3: “The authors seem to have missed the spirit of my critique: to say "linear readout is performed in motor cortex" is an over-interpretation of what their model can show.”

Thank you for your comments. It's important to note that the conclusions we draw are speculative and not definitive. We use terms like "suggest" to reflect this uncertainty. To further emphasize the conjectural nature of our conclusions, we have deliberately moderated our tone.

The question of whether behaviorally-relevant signals can be accessed by downstream brain regions hinges on the debate over whether the brain employs a strategy of filtering before decoding. If the brain employs such a strategy, the brain can probably access these signals. In our view, it is likely that the brain utilizes this strategy.

Given the existence of behaviorally relevant signals, it is reasonable to assume that the brain has intrinsic mechanisms to differentiate between relevant and irrelevant signals. There is growing evidence suggesting that the brain utilizes various mechanisms, such as attention and specialized filtering, to suppress irrelevant signals and enhance relevant signals [1-3]. Therefore, it is plausible that the brain filters before decoding, thereby effectively accessing behaviorally relevant signals.

Regarding the question of whether the brain employs linear readout, given the limitations of current observational methods and our incomplete understanding of brain mechanisms, it is challenging to ascertain whether the brain employs a linear readout. In many cortical areas, linear decoders have proven to be sufficiently accurate. Consequently, numerous studies [4, 5, 6], including the one you referenced [4], directly employ linear decoders to extract information and formulate conclusions based on the decoding results. Contrary to these approaches, our research has compared the performance of linear and nonlinear decoders on behaviorally relevant signals and found their decoding performance is comparable. Considering both the decoding accuracy and model complexity, our results suggest that the motor cortex may utilize linear readout to decode information from relevant signals. Given the current technological limitations, we consider it reasonable to analyze collected data to speculate on the potential workings of the brain, an approach that many studies have also embraced [7-10]. For instance, a study [7] deduces strategies the brain might employ to overcome noise by analyzing the structure of recorded data and decoding outcomes for new stimuli.

Thank you for your valuable feedback.

(1) Sreenivasan, Sameet, and Ila Fiete. "Grid cells generate an analog error-correcting code for singularly precise neural computation." Nature neuroscience 14.10 (2011): 1330-1337.

(2) Schneider, David M., Janani Sundararajan, and Richard Mooney. "A cortical filter that learns to suppress the acoustic consequences of movement." Nature 561.7723 (2018): 391-395.

(3) Nakajima, Miho, L. Ian Schmitt, and Michael M. Halassa. "Prefrontal cortex regulates sensory filtering through a basal ganglia-to-thalamus pathway." Neuron 103.3 (2019): 445-458.

(4) Jurewicz, Katarzyna, et al. "Irrational choices via a curvilinear representational geometry for value." bioRxiv (2022): 2022-03.

(5) Hong, Ha, et al. "Explicit information for category-orthogonal object properties increases along the ventral stream." Nature neuroscience 19.4 (2016): 613-622.

(6) Chang, Le, and Doris Y. Tsao. "The code for facial identity in the primate brain." Cell 169.6 (2017): 1013-1028.

(7) Ganmor, Elad, Ronen Segev, and Elad Schneidman. "A thesaurus for a neural population code." Elife 4 (2015): e06134.

(8) Churchland, Mark M., et al. "Neural population dynamics during reaching." Nature 487.7405 (2012): 51-56.

(9) Gallego, Juan A., et al. "Cortical population activity within a preserved neural manifold underlies multiple motor behaviors." Nature communications 9.1 (2018): 4233.

(10) Gallego, Juan A., et al. "Long-term stability of cortical population dynamics underlying consistent behavior." Nature neuroscience 23.2 (2020): 260-270.

Q4: “Agreeing with my critique is not sufficient; please provide the data or simulations that provides the context for the reference in the fano factor. I believe my critique is still valid.”

Thank you for your comments. As we previously replied, Churchland's research examines the variability of neural signals across different stages, including the preparation and execution phases, as well as before and after the target appears. Our study, however, focuses exclusively on the movement execution phase. Consequently, we are unable to produce comparative displays similar to those in his research. Intuitively, one might expect that the variability of behaviorally relevant signals would be lower; however, since no prior studies have accurately extracted such signals, the specific FF values of behaviorally relevant signals remain unknown. Therefore, presenting these values is meaningful, and can provide a reference for future research. While we cannot compare FF across different stages, we can numerically compare the values to the Poisson count process. An FF of 1 indicates a Poisson firing process, and our experimental data reveals that most neurons have an FF less than 1, indicating that the variance in firing counts is below the mean. Thank you for your valuable feedback.

To Reviewer #4

Q1: “Overall, studying neural computations that are behaviorally relevant or not is an important problem, which several previous studies have explored (for example PSID in (Sani et al. 2021), TNDM in (Hurwitz et al. 2021), TAME-GP in (Balzani et al. 2023), pi-VAE in (Zhou and Wei 2020), and dPCA in (Kobak et al. 2016), etc). However, this manuscript does not properly put their work in the context of such prior works. For example, the abstract states "One solution is to accurately separate behaviorally-relevant and irrelevant signals, but this approach remains elusive", which is not the case given that these prior works have done that. The same is true for various claims in the main text, for example "Furthermore, we found that the dimensionality of primary subspace of raw signals (26, 64, and 45 for datasets A, B, and C) is significantly higher than that of behaviorally-relevant signals (7, 13, and 9), indicating that using raw signals to estimate the neural dimensionality of behaviors leads to an overestimation" (line 321). This finding was presented in (Sani et al. 2021) and (Hurwitz et al. 2021), which is not clarified here. This issue of putting the work in context has been brought up by other reviewers previously but seems to remain largely unaddressed. The introduction is inaccurate also in that it mixes up methods that were designed for separation of behaviorally relevant information with those that are unsupervised and do not aim to do so (e.g., LFADS). The introduction should be significantly revised to explicitly discuss prior models/works that specifically formulated this behavior separation and what these prior studies found, and how this study differs.”

Thank you for your comments. Our statement about “One solution is to accurately separate behaviorally-relevant and irrelevant signals, but this approach remains elusive” is accurate. To our best knowledge, there is no prior works to do this work--- separating accurate behaviorally relevant neural signals at both single-neuron and single-trial resolution. The works you mentioned have not explicitly proposed extracting behaviorally relevant signals, nor have they identified and addressed the key challenges of extracting relevant signals, namely determining the optimal degree of similarity between the generated relevant signals and raw signals. Those works focus on the latent neural dynamics, rather than signal level.

To clearly set apart d-VAE from other models, we have framed the extraction of behaviorally relevant signals as the following mathematical optimization problem:minxrE(xr,x)+R(xr)

where 𝒙𝒓 denotes generated behaviorally-relevant signals, 𝒙 denotes raw noisy signals, 𝐸(⋅,⋅) demotes reconstruction loss, and 𝑅(⋅) denotes regularization loss. It is important to note that while both d-VAE and TNDM employ reconstruction loss, relying solely on this term is insufficient for determining the optimal degree of similarity between the generated and raw noisy signals. The key to accurately extracting behaviorally relevant signals lies in leveraging prior knowledge about these signals to determine the optimal similarity degree, encapsulated by 𝑅(𝒙𝒓). All the works you mentioned did not have the key part 𝑅(𝒙𝒓).

Regarding the dimensionality estimation, the dimensionality of neural manifolds quantifies the degrees of freedom required to describe population activity without significant information loss.

There are two differences between our work and PSID and TNDM.

First, the dimensions they refer to are fundamentally different from ours. The dimensionality we describe pertains to a linear subspace, where a neural dimension or neural mode or principal component basis, u∈RN , with N representing the number of neurons. However, the vector length of a neural mode of PSID and our approach differs; PSID requires concatenating multiple time steps T, essentially making u∈RNT , TNDM, on the other hand, involves nonlinear dimensionality reduction, which is different from linear dimensionality reduction.

Second, we estimate neural dimensionality by explaining the variance of neural signals, whereas PSID and TNDM determine dimensionality through decoding performance saturation. It is important to note that the dimensionality at which decoding performance saturates may not accurately reflect the true dimensionality of neural manifolds, as some dimensions may contain redundant information that does not enhance decoding performance.

We acknowledge that while LFADS can generate signals that contain some behavioral information, it was not specifically designed to do so. Following your suggestion, we have removed this reference from the Introduction.

Thank you for your valuable feedback.

Q2: “Claims about linearity of "motor cortex" readout are not supported by results yet stated even in the abstract. Instead, what the results support is that for decoding behavior from the output of the dVAE model -- that is trained specifically to have a linear behavior readout from its embedding -- a nonlinear readout does not help. This result can be biased by the very construction of the dVAE's loss that encourages a linear readout/decoding from embeddings, and thus does not imply a finding about motor cortex.”

Thank you for your comments. We respectfully disagree with the notion that the ability of relevant signals to be linearly decoded is due to constraints that allow embedding to be linearly decoded. Embedding involves reorganizing or transforming the structure of original signals, and they can be linearly decoded does not mean the corresponding signals can be decoded linearly.

Let's clarify this with three intuitive examples:

Example 1: Image denoising is a well-established field. Whether employing supervised or blind denoising methods [1, 2], both can effectively recover the original image. This denoising process closely resembles the extraction of behaviorally relevant signals from raw signals. Consider if noisy images are not amenable to linear decoding (classification); would removing the noise enable linear decoding? The answer is no. Typically, the noise in images captured under normal conditions is minimal, yet even the clear images remain challenging to decode linearly.

Example 2: Consider the task of face recognition, where face images are set against various backgrounds, in this context, the pixels representing the face corresponds to relevant signals, while the background pixels are considered irrelevant. Suppose a network is capable of extracting the face pixels and the resulting embedding can be linearly decoded. Can the face pixels themselves be linearly decoded? The answer is no. If linear decoding of face pixels were feasible, the challenging task of face recognition could be easily resolved by merely extracting the face from the background and training a linear classifier.

Example 3: In the MNIST dataset, the background is uniformly black, and its impact is minimal. However, linear SVM classifiers used directly on the original pixels significantly underperform compared to non-linear SVMs.

In summary, embedding involves reorganizing the structure of the original signals through a feature transformation function. However, the reconstruction process can recover the structure of the original signals from the embedding. The fact that the structure of the embedding can be linearly decoded does not imply that the structure of the original signals can be linearly decoded in the same way. It is inappropriate to focus on the compression process without equally considering the reconstruction process.

Thank you for your valuable feedback.

(1) Mao, Xiao-Jiao, Chunhua Shen, and Yu-Bin Yang. "Image restoration using convolutional auto-encoders with symmetric skip connections." arXiv preprint arXiv:1606.08921 (2016).

(2) Lehtinen, Jaakko, et al. "Noise2Noise: Learning image restoration without clean data." International Conference on Machine Learning. International Machine Learning Society, 2018.

Q3: “Related to the above, it is unclear what the manuscript means by readout from motor cortex. A clearer definition of "readout" (a mapping from what to what?) in general is needed. The mapping that the linearity/nonlinearity claims refer to is from the *inferred* behaviorally relevant neural signals, which themselves are inferred nonlinearly using the VAE. This should be explicitly clarified in all claims, i.e., that only the mapping from distilled signals to behavior is linear, not the whole mapping from neural data to behavior. Again, to say the readout from motor cortex is linear is not supported, including in the abstract.”

Thank you for your comments. We have revised the manuscript to make it more clearly. Thank you for your valuable feedback.

Q4: “Claims about individual neurons are also confounded. The d-VAE distilling processing is a population level embedding so the individual distilled neurons are not obtainable on their own without using the population data. This population level approach also raises the possibility that information can leak from one neuron to another during distillation, which is indeed what the authors hope would recover true information about individual neurons that wasn't there in the recording (the pixel denoising example). The authors acknowledge the possibility that information could leak to a neuron that didn't truly have that information and try to rule it out to some extent with some simulations and by comparing the distilled behaviorally relevant signals to the original neural signals. But ultimately, the distilled signals are different enough from the original signals to substantially improve decoding of low information neurons, and one cannot be sure if all of the information in distilled signals from any individual neuron truly belongs to that neuron. It is still quite likely that some of the improved behavior prediction of the distilled version of low-information neurons is due to leakage of behaviorally relevant information from other neurons, not the former's inherent behavioral information. This should be explicitly acknowledged in the manuscript.”

Thank you for your comments. We value your insights regarding the mixing process. However, we are confident in the robustness of our conclusions. We respectfully disagree with the notion that the small R2 values containing significant information are primarily due to leakage, and we base our disagreement on four key reasons.

(1) Neural reconstruction performance is a reliable and valid criterion.

The purpose of latent variable models is to explain neuronal activity as much as possible. Given the fact that the ground truth of behaviorally-relevant signals, the latent variables, and the generative model is unknow, it becomes evident that the only reliable reference at the signal level is the raw signals. A crucial criterion for evaluating the reliability of latent variable models (including latent variables and generated relevant signals) is their capability to effectively explain the raw signals [1]. Consequently, we firmly maintain the belief that if the generated signals closely resemble the raw signals to the greatest extent possible, in accordance with an equivalence principle, we can claim that these obtained signals faithfully retain the inherent properties of single neurons.

Reviewer #4 appears to focus on the compression (mixing) process without giving equal consideration to the reconstruction (de-mixing) process. Numerous studies have demonstrated that deep autoencoders can reconstruct the original signal very effectively. For example, in the field of image denoising, autoencoders are capable of accurately restoring the original image [2, 3]. If one persistently focuses on the fact of mixing and ignores the reconstruction （demix） process, even if the only criterion that we can rely on at the signal level is high, one still won't acknowledge it. If this were the case, many problems would become unsolvable. For instance, a fundamental criterion for latent variable models is their ability to explain the original data. If the ground truth of the latent variables remains unknown and the reconstruction criterion is disregarded, how can we validate the effectiveness of the model, the validity of the latent variables, or ensure that findings related to latent variables are not merely by-products of the model? Therefore, we disagree with the aforementioned notion. We believe that as long as the reconstruction performance is satisfactory, the extracted signals have successfully retained the characteristics of individual neurons.

In our paper, we have shown in various ways that our generated signals sufficiently resemble the raw signals, including visualizing neuronal activity (Fig. 2m, Fig. 3i, and Fig. S5), achieving the highest performance among competitors (Fig. 2d, h, l), and conducting control analyses. Therefore, we believe our results are reliable.

(1) Cunningham, J.P. and Yu, B.M., 2014. Dimensionality reduction for large-scale neural recordings. Nature neuroscience, 17(11), pp.1500-1509.

(2) Mao, Xiao-Jiao, Chunhua Shen, and Yu-Bin Yang. "Image restoration using convolutional auto-encoders with symmetric skip connections." arXiv preprint arXiv:1606.08921 (2016).

(3) Lehtinen, Jaakko, et al. "Noise2Noise: Learning image restoration without clean data." International Conference on Machine Learning. International Machine Learning Society, 2018.

(2) There is no reason for d-VAE to add signals that do not exist in the original signals.

(1) Adding signals that does not exist in the small R2 neurons would decrease the reconstruction performance. This is because if the added signals contain significant information, they will not resemble the irrelevant signals which contain no information, and thus, the generated signals will not resemble the raw signals. The model optimizes towards reducing the reconstruction loss, and this scenario deviates from the model's optimization direction. It is worth mentioning that when the model only has reconstruction loss without the interference of decoding loss, we believe that information leakage does not happen. Because the model can only be optimized in a direction that is similar to the raw signals; adding non-existent signals to the generated signals would increase the reconstruction loss, which is contrary to the objective of optimization.

(2) Information carried by these additional signals is redundant for larger R2 neurons, thus they do not introduce new information that can enhance the decoding performance of the neural population, which does not benefit the decoding loss.

Based on these two points, we believe the model would not perform such counterproductive and harmful operations.

(3) The criterion that irrelevant signals should contain minimal information can effectively rule out the leakage scenario.

The criterion that irrelevant signals should contain minimal information is very important, but it seems that reviewer #4 has continuously overlooked their significance. If the model's reconstruction is insufficient, or if additional information is added (which we do not believe will happen), the residuals would decode a large amount of information, and this criterion would exclude selecting such signals. To clarify, if we assume that x, y, and z denote the raw, relevant, and irrelevant signals of smaller R2 neurons, with x=y+z, and the extracted relevant signals become y+m, the irrelevant signals become z-m in this case. Consequently, the irrelevant signals contain a significant amount of information.

We presented the decoding R2 for irrelevant signals in real datasets under three distillation scenarios: a bias towards reconstruction (alpha=0, an extreme case where the model only has reconstruction loss without decoding loss), a balanced trade-off, and a bias towards decoding (alpha=0.9), as detailed in Table 1. If significant information from small R2 neurons leaks from large R2 neurons, the irrelevant signals should contain a large amount of information. However, our results indicate that the irrelevant signals contain only minimal information, and their performance closely resembles that of the model training solely with reconstruction loss, showing no significant differences (P > 0.05, Wilcoxon rank-sum test). When the model leans towards decoding, some useful information will be left in the residuals, and irrelevant signals will contain a substantial amount of information, as observed in Table 1, alpha=0.9. Therefore, we will not choose these signals for analysis.

In conclusion, the criterion that irrelevant signals should contain minimal information is a very effective measure to exclude undesirable signals.

**Author response table 1. sa4table1:** Decoding R2 of irrelevant signals.

	Dataset A	Dataset B	Dataset C
Alpha = 0	0.065+-0.027	0.098+-0.037	0.071+-0.034
Selected alpha	0.105+-0.032	0.067+-0.031	0.095+-0.037
Alpha = 0.9	0.220+-0.045	0.106+-0.044	0.182+-0.056

(4) Synthetic experiments can effectively rule out the leakage scenario.

In the absence of ground truth data, synthetic experiments serve as an effective method for validating models and are commonly employed [1-3].

Our experimental results demonstrate that d-VAE can effectively extract neural signals that more closely resemble actual behaviorally relevant signals (Fig. S2g). If there were information leakage, it would decrease the similarity to the ground truth signals, hence we have ruled out this possibility. Moreover, in synthetic experiments with small R2 neurons (Fig. S10), results also demonstrate that our model could make these neurons more closely resemble ground truth relevant signals and recover their information.

In summary, synthetic experiments strongly demonstrate that our model can recover obscured neuronal information, rather than adding signals that do not exist.

(1) Pnevmatikakis, Eftychios A., et al. "Simultaneous denoising, deconvolution, and demixing of calcium imaging data." Neuron 89.2 (2016): 285-299.

(2) Schneider, Steffen, Jin Hwa Lee, and Mackenzie Weygandt Mathis. "Learnable latent embeddings for joint behavioural and neural analysis." Nature 617.7960 (2023): 360-368.

(3) Zhou, Ding, and Xue-Xin Wei. "Learning identifiable and interpretable latent models of high-dimensional neural activity using pi-VAE." Advances in Neural Information Processing Systems 33 (2020): 7234-7247.

Based on these four points, we are confident in the reliability of our results. If Reviewer #4 considers these points insufficient, we would highly appreciate it if specific concerns regarding any of these aspects could be detailed.

Thank you for your valuable feedback.

Q5: “Given the nuances involved in appropriate comparisons across methods and since two of the datasets are public, the authors should provide their complete code (not just the dVAE method code), including the code for data loading, data preprocessing, model fitting and model evaluation for all methods and public datasets. This will alleviate concerns and allow readers to confirm conclusions (e.g., figure 2) for themselves down the line.”

Thanks for your suggestion.

Our codes are now available on GitHub at https://github.com/eric0li/d-VAE. Thank you for your valuable feedback.

Q6: “Related to (1) above, the authors should explore the results if the affine network h(.) (from embedding to behavior) was replaced with a nonlinear ANN. Perhaps linear decoders would no longer be as close to nonlinear decoders. Regardless, the claim of linearity should be revised as described in (1) and (2) above, and all caveats should be discussed.”

Thank you for your suggestion. We appreciate your feasible proposal that can be empirically tested. Following your suggestion, we have replaced the decoding of the latent variable z to behavior y with a nonlinear neural network, specifically a neural network with a single hidden layer. The modified model is termed d-VAE2. We applied the d-VAE2 to the real data, and selected the optimal alpha through the validation set. As shown in Table 1, results demonstrate that the performance of KF and ANN remains comparable. Therefore, the capacity to linearly decode behaviorally relevant signals does not stem from the linear decoding of embeddings.

**Author response table 2. sa4table2:** Decoding R2 of behaviorally relevant signals obtained by d-VAE2.

	Dataset A	Dataset B	Dataset C
KF	0.706+-0.016	0.704+-0.039	0.860+-0.012
ANN	0.752+-0.010	0.738+-0.033	0.870+-0.009

Additionally, it is worth noting that this approach is uncommon and is considered somewhat inappropriate according to the Information Bottleneck theory [1]. According to the Information Bottleneck theory, information is progressively compressed in multilayer neural networks, discarding what is irrelevant to the output and retaining what is relevant. This means that as the number of layers increases, the mutual information between each layer's embedding and the model input gradually decreases, while the mutual information between each layer's embedding and the model output gradually increases. For the decoding part, if the embeddings that is not closest to the output (behaviors) is used, then these embeddings might contain behaviorally irrelevant signals. Using these embeddings to generate behaviorally relevant signals could lead to the inclusion of irrelevant signals in the behaviorally relevant signals.

To demonstrate the above statement, we conducted experiments on the synthetic data. As shown in Table 2, we present the performance (neural R2 between the generated signals and the ground truth signals) of both models at several alpha values around the optimal alpha of dVAE (alpha=0.9) selected by the validation set. The experimental results show that at the same alpha value, the performance of d-VAE2 is consistently inferior to that of d-VAE, and d-VAE2 requires a higher alpha value to achieve performance comparable to d-VAE, and the best performance of d-VAE2 is inferior to that of d-VAE.

**Author response table 3. sa4table3:** Neural R2 between generated signals and real behaviorally relevant signals.

Alpha	0.7	0.8	0.9	1	2	3
d-	0.720+-	0.716+-	0.712+-	0.730+-	0.736+-	0.713+-
VAE	0.014	0.028	0.023	0.021	0.009	0.013
d-	0.689+-	0.693+-	0.703+-	0.720+-	0.727+-	0.679+-
VAE2	0.033	0.051	0.006	0.019	0.015	0.027

Thank you for your valuable feedback.

(1) Shwartz-Ziv, Ravid, and Naftali Tishby. "Opening the black box of deep neural networks via information." arXiv preprint arXiv:1703.00810 (2017).

Q7: “The beginning of the section on the "smaller R2 neurons" should clearly define what R2 is being discussed. Based on the response to previous reviewers, this R2 "signifies the proportion of neuronal activity variance explained by the linear encoding model, calculated using raw signals". This should be mentioned and made clear in the main text whenever this R2 is referred to.”

Thank you for your suggestion. We have made the modifications in the main text. Thank you for your valuable feedback.

Q8: “Various terms require clear definitions. The authors sometimes use vague terminology (e.g., "useless") without a clear definition. Similarly, discussions regarding dimensionality could benefit from more precise definitions. How is neural dimensionality defined? For example, how is "neural dimensionality of specific behaviors" (line 590) defined? Related to this, I agree with Reviewer 2 that a clear definition of irrelevant should be mentioned that clarifies that relevance is roughly taken as "correlated or predictive with a fixed time lag". The analyses do not explore relevance with arbitrary time lags between neural and behavior data.”

Thanks for your suggestion. We have removed the “useless” statements and have revised the statement of “the neural dimensionality of specific behaviors” in our revised manuscripts.

Regarding the use of fixed temporal lags, we followed the same practice as papers related to the dataset we use, which assume fixed temporal lags [1-3]. Furthermore, many studies in the motor cortex similarly use fixed temporal lags [4-6]. To clarify the definition, we have revised the definition in our manuscript. For details, please refer to the response to Q2 of reviewer #2 and our revised manuscript. We believe our definition is clearly articulated.

Thank you for your valuable feedback.

(1) Wang, Fang, et al. "Quantized attention-gated kernel reinforcement learning for brain– machine interface decoding." IEEE transactions on neural networks and learning systems 28.4 (2015): 873-886.

(2) Dyer, Eva L., et al. "A cryptography-based approach for movement decoding." Nature biomedical engineering 1.12 (2017): 967-976.

(3) Ahmadi, Nur, Timothy G. Constandinou, and Christos-Savvas Bouganis. "Robust and accurate decoding of hand kinematics from entire spiking activity using deep learning." Journal of Neural Engineering 18.2 (2021): 026011.

(4) Churchland, Mark M., et al. "Neural population dynamics during reaching." Nature 487.7405 (2012): 51-56.

(5) Kaufman, Matthew T., et al. "Cortical activity in the null space: permitting preparation without movement." Nature neuroscience 17.3 (2014): 440-448.

(6) Elsayed, Gamaleldin F., et al. "Reorganization between preparatory and movement population responses in motor cortex." Nature communications 7.1 (2016): 13239.

Q9: “CEBRA itself doesn't provide a neural reconstruction from its embeddings, but one could obtain one via a regression from extracted CEBRA embeddings to neural data. In addition to decoding results of CEBRA (figure S3), the neural reconstruction of CEBRA should be computed and CEBRA should be added to Figure 2 to see how the behaviorally relevant and irrelevant signals from CEBRA compare to other methods.”

Thank you for your question. Modifying CEBRA is beyond the scope of our work. As CEBRA is not a generative model, it cannot obtain behaviorally relevant and irrelevant signals, and therefore it lacks the results presented in Fig. 2. To avoid the same confusion encountered by reviewers #3 and #4 among our readers, we have opted to exclude the comparison with CEBRA. It is crucial to note, as previously stated, that our assessment of decoding capabilities has been benchmarked against the performance of the ANN on raw signals, which almost represents the upper limit of performance. Consequently, omitting CEBRA does not affect our conclusions.

Thank you for your valuable feedback.

Q10: “Line 923: "The optimal hyperparameter is selected based on the lowest averaged loss of five-fold training data." => why is this explained specifically under CEBRA? Isn't the same criteria used for hyperparameters of other methods? If so, clarify.”

Thank you for your question. The hyperparameter selection for CEBRA follows the practice of the original CEBRA paper. The hyperparameter selection for generative models is detailed in the Section “The strategy for selecting effective behaviorally-relevant signals”. Thank you for your valuable feedback.